# Receptor-specific interactome as a hub for rapid cue-induced selective translation in axons

Max Koppers[1], Roberta Cagnetta[1†], Toshiaki Shigeoka[1†], Lucia CS Wunderlich[2], Pedro Vallejo-Ramirez[2], Julie Qiaojin Lin[1], Sixian Zhao[1], Maximilian AH Jakobs[1], Asha Dwivedy[1,2], Michael S Minett[1], Anaïs Bellon[1‡], Clemens F Kaminski[2], William A Harris[1], John G Flanagan[3], Christine E Holt[1]*

[1]Department of Physiology, Development and Neuroscience, University of Cambridge, Cambridge, United Kingdom; [2]Department of Chemical Engineering and Biotechnology, University of Cambridge, Cambridge, United Kingdom; [3]Department of Cell Biology, Harvard Medical School, Boston, United States

*For correspondence:
ceh33@cam.ac.uk

†These authors contributed equally to this work

Present address: ‡Aix Marseille University, INSERM, Institute de Neurobiologie de la Méditerranée, Marseille, France

Competing interests: The authors declare that no competing interests exist.

## Abstract

Extrinsic cues trigger the local translation of specific mRNAs in growing axons via cell surface receptors. The coupling of ribosomes to receptors has been proposed as a mechanism linking signals to local translation but it is not known how broadly this mechanism operates, nor whether it can selectively regulate mRNA translation. We report that receptor-ribosome coupling is employed by multiple guidance cue receptors and this interaction is mRNA-dependent. We find that different receptors associate with distinct sets of mRNAs and RNA-binding proteins. Cue stimulation of growing *Xenopus* retinal ganglion cell axons induces rapid dissociation of ribosomes from receptors and the selective translation of receptor-specific mRNAs. Further, we show that receptor-ribosome dissociation and cue-induced selective translation are inhibited by co-exposure to translation-repressive cues, suggesting a novel mode of signal integration. Our findings reveal receptor-specific interactomes and suggest a generalizable model for cue-selective control of the local proteome.

## Introduction

mRNA localization and local translation are major determinants of the local proteome (*Zappulo et al., 2017*). This seems particularly important for morphologically complex cells such as neurons, where the axonal sub-compartment and its growing tip, the growth cone, often far away from the cell body, rapidly perform specialized functions (*Holt and Schuman, 2013*). During neuronal wiring, specific interactions between extrinsic cues and receptors mediate guidance of axons to their proper target area and axon branching in this area (*Stoeckli, 2018*; *Manitt et al., 2009*; *Marshak et al., 2007*; *Cioni et al., 2013*). The rapid axonal responses to several guidance cues require local protein synthesis (*Jung et al., 2012*; *Campbell and Holt, 2001*). For example, attractive guidance cues, such as Netrin-1, trigger axonal translation of mRNAs encoding proteins that facilitate actin assembly, whereas repulsive cues trigger the local synthesis of cytoskeletal proteins involved in actin disassembly (*Leung et al., 2006*; *Wu et al., 2005*; *Piper et al., 2006*). This cue-specific mode of translation enables growth cones to steer differentially – towards or away – from the source of such cues (*Lin and Holt, 2007*; *Lin and Holt, 2008*). Unbiased detection of newly synthesized proteins in the axon compartment has revealed further complexity showing that different guidance cues stimulate the regulation of distinct signature sets of >100 axonal nascent proteins within just 5 min, many of which are not cytoskeletal-related (*Leung et al., 2006*; *Yao et al., 2006*; *Wu et al., 2005*; *Cagnetta et al., 2018*; *Cioni et al., 2018*). Several mechanisms are known to

control different aspects of axonal translation, including microRNA regulation (*Bellon et al., 2017*), mRNA modification (*Yu et al., 2018*), modulation of the phosphorylation of eukaryotic initiation factors (*Cagnetta et al., 2019*), RNA-binding protein (RBP) phosphorylation (*Sasaki et al., 2010*; *Lepelletier et al., 2017*; *Hüttelmaier et al., 2005*) and receptor-ribosome coupling (*Tcherkezian et al., 2010*). The latter is a particularly direct and attractive mechanism to link cue-specific signalling to differential mRNA translation. However, this mechanism has been shown only for the Netrin-1 receptor, deleted in colorectal cancer (DCC), in commissural axon growth cones and HEK293 cells (*Tcherkezian et al., 2010*). It is unknown whether receptor-ribosome coupling is a widespread mechanism used by different receptors and in different cell types, and whether it regulates selective local translation.

Here, we show in the axonal growth cones of retinal ganglion cells (RGCs) that receptor-ribosome coupling is used by several different guidance receptors known to trigger local protein synthesis (DCC, Neuropilin-1 and Robo2, but not EphB2), indicative of a common mechanism. Interestingly, the receptor-ribosome interaction is mRNA-dependent and immunoprecipitation (IP) reveals that distinct receptors associate with specific RNA-binding proteins (RBPs) and subsets of mRNAs. Upon cue-stimulation, ribosomes dissociate from their receptors within 2 min and receptor-specific mRNAs are selectively translated. We also find that co-stimulation with EphrinA1 blocks the Netrin-1-induced DCC receptor-ribosome dissociation and selective translation in axons, suggesting a new regulatory mechanism for integrating different signals. Together, this study provides evidence that receptor-ribosome coupling is a common mechanism across different receptors and cell types, and suggests that receptor-specific interactomes act as a hub to regulate the localized and selective cue-induced mRNA translation.

## Results

### Multiple guidance cue receptors interact with ribosomes

In retinal axons, Netrin-1 and Sema3A mediate growth cone steering and branching (*Campbell and Holt, 2001*; *Manitt et al., 2009*; *Campbell et al., 2001*). Specifically, the rapid chemotropic responses to Netrin-1 and Sema3A are mediated, at least in part, by local translation (*Campbell and Holt, 2001*). The Netrin-1 receptor, DCC, was previously reported to associate with ribosomes in spinal commissural axon growth cones (*Tcherkezian et al., 2010*). We first asked whether the interaction of DCC with ribosomes is conserved in a different system and cell type, and explored the possibility that the Sema3A receptor, Neuropilin-1 (Nrp1), also interacts with ribosomes in this system. To do this, we performed immunoprecipitation (IP) of endogenous DCC and Nrp1 from *Xenopus laevis* embryonic brains and eyes followed by mass-spectrometry (LC-MS/MS) analysis of eluted samples. Each IP was performed in triplicate and after raw data processing using MaxQuant software, we determined statistically significant interactors of DCC and Nrp1 compared to an IgG control pull-down using label-free (LFQ) intensities and Perseus software analysis (*Figure 1A*). Gene-ontology (GO) enrichment analysis revealed that 'structural constituent of ribosomes' appeared as the most prominently enriched category in both DCC and Nrp1 pulldowns, indicating that both receptors can interact with ribosomal proteins (*Figure 1B*). Specifically, 75 out of 79 ribosomal proteins (94.9%) were detected in the DCC and Nrp1 pulldowns. Of these, 51 and 33 RPs were identified as statistically enriched interactors for Nrp1 and DCC, respectively, compared to IgG control pulldowns. There was no bias towards small or large ribosomal subunit proteins (*Figure 1A*, red dots). The GO analysis also revealed the presence of other groups shared between the receptors, such as 'vesicle-mediated transport' (*Figure 1B*). Interestingly, some categories of proteins were enriched for only one of the receptors, for example the 'phosphoprotein phosphatase activity' GO term was significantly enriched only in the DCC pulldown and the 'barbed-end actin filament capping' GO term was enriched only in the Nrp1 pulldown (*Figure 1B*). To confirm the interaction between receptors and ribosomal proteins, we performed Western blot (WB) analysis after IP and validated that both DCC and Nrp1 interact with small (40S) and large (60S) ribosomal subunit proteins (*Figure 1C–D*). These interactions appear to be conserved, as endogenous IP from the human neuronal cell line SH-SY5Y, which expresses both DCC and Nrp1, also shows ribosomal protein co-precipitation after pulldown of the endogenous receptor (*Figure 1—figure supplement 1A–B*).

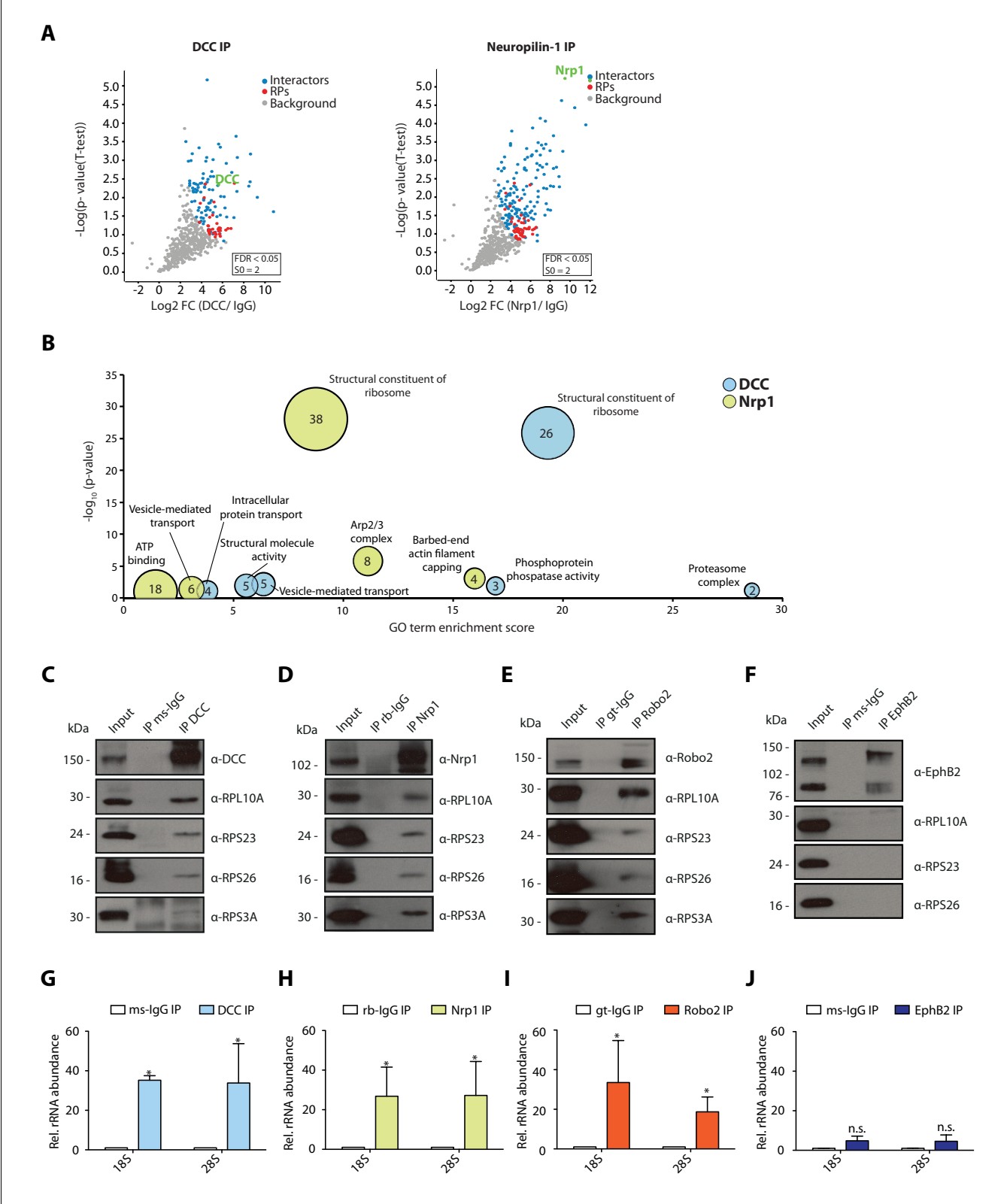

**Figure 1.** Multiple guidance cue receptors interact with ribosomes. (**A**) Volcano plots showing statistically enriched proteins in DCC-IP and Nrp1-IP samples identified by permutation-based FDR-corrected t-test based on three biological replicates. The LFQ intensity of the DCC or Nrp1 pulldowns over IgG pulldowns are plotted against the -log10 p-value. FDR < 0.05; S0 = 2. (**B**) Gene enrichment analysis of statistically enriched proteins in the DCC and Nrp1 pulldown samples. The values in each circle denotes protein count. (**C–F**) Western blot validation of RP co-immunoprecipitation with

*Figure 1 continued on next page*

*Figure 1 continued*

DCC, Nrp1 and Robo2 but not with EphB2. Each Western blot was repeated 2 to 4 times, representative images are shown. (**G–J**) Relative 18S and 28S ribosomal RNA abundance after control (IgG) pulldown or receptors pulldowns shows enrichment of rRNA in DCC, Nrp1, and Robo2 but not EphB2 pulldowns (unpaired two-tailed t-test; three biological replicates). Bars indicate means, error bars indicate standard deviation; *p<0.05.

The online version of this article includes the following figure supplement(s) for figure 1:

**Figure supplement 1.** Multiple guidance cue receptors interact with ribosomes in SH-SY5Y cells.

In addition to DCC and Nrp1, Roundabout 2 (Robo2) triggers local protein synthesis after binding to the guidance cue Slit2 (*Piper et al., 2006*). Therefore, we asked whether Robo2 also interacts with ribosomal proteins. WB after IP from *Xenopus* embryonic brains and eyes or SH-SY5Y cells showed that Robo2 also interacts with ribosomal proteins of both subunits (*Figure 1E*, *Figure 1—figure supplement 1C*). We then looked at EphB2, as growth cone collapse mediated by EphrinB, the ligand for this receptor, is not mediated by local protein synthesis (*Mann et al., 2003*). In this case, we could not detect co-IP of ribosomal proteins with EphB2 in *Xenopus* embryonic brains and eyes, indicating that not all guidance receptors interact with ribosomal proteins (*Figure 1F*), and suggesting that only receptors that require local protein synthesis for their action on growth cones are coupled to ribosomes.

To confirm that receptors bind to ribosomes or ribosomal subunits and not free ribosomal proteins, we isolated RNA after IP and performed quantitative-RT-PCR (qPCR) for 18S (40S small ribosomal subunit) and 28S (60S large ribosomal subunit) ribosomal RNA (rRNA), which should be present only in intact ribosomal subunits in the cytoplasm. Consistent with the WB results, DCC, Nrp1 and Robo2, but not EphB2, exhibit a significant enrichment of both 18S rRNA and 28S rRNA compared to an IgG control pulldown in *Xenopus* brains (*Figure 1G–J*), and in SH-SY5Y cells in the case of DCC and Nrp1 (*Figure 1—figure supplement 1D–E*). Collectively, these findings reveal that multiple receptors known to trigger local protein synthesis can associate with ribosomal subunits.

## Guidance cue receptors associate with ribosomes in a mRNA-dependent manner

We next examined the co-sedimentation profiles of DCC and Nrp1 in *Xenopus* embryonic brains and eyes after sucrose gradient purification of ribosomes in order to see if the receptors were mostly associated with ribosomal subunits, monosomes or polysomes. Consistent with previous findings (*Tcherkezian et al., 2010*), DCC was prominent in 40S, 60S and 80S fractions but not in polysomal fractions (*Figure 2—figure supplement 1A*). Nrp1, however, was found in 40S, 60S and 80S fractions, as well as in polysomal fractions (*Figure 2—figure supplement 1A*), suggesting a possibly different association mechanism or a different translational status of the receptor-bound ribosomes. Both DCC and Nrp1 were also present in ribosome-free fractions indicating that not all receptor molecules are associated with ribosomes (*Figure 2—figure supplement 1A,C*). EDTA treatment, which dissociates the monosomes/polysomes into separate ribosomal subunits (*Simsek et al., 2017*), shifted both DCC and Nrp1 to lighter fractions, supporting a valid association with ribosomes (*Figure 2—figure supplement 1B,C*).

We used qPCR to investigate this association further. When IP samples were treated with EDTA before elution, the enrichment of 18S and 28S rRNA after receptor pulldown was significantly decreased for both DCC and Nrp1 (*Figure 2A*). A possible explanation for this decrease is that DCC and Nrp1 interact mainly with 80S ribosomes (*Tcherkezian et al., 2010*). Another possibility is that the binding of ribosomes to receptors is mRNA-dependent. To test the latter hypothesis, we treated the receptor pulldown samples with RNase A/T1, which digests mRNAs and releases any factors bound to ribosomes via mRNA (*Simsek et al., 2017*). The concentration of RNase A/T1 used here largely preserves the integrity of ribosomes, as evidenced by the co-sedimentation profiles that show successful conversion of polysomes into monosomes, increasing the monosomal (80S) peak (*Figure 2—figure supplement 1D*), though we cannot exclude that it may still partially cleave rRNA. The significant decrease in the co-sedimentation of 18S and 28S rRNA with receptors in these conditions suggests that mRNA is important for the association of 80S ribosomes with receptors (*Figure 2A*). Consistent with these results, Western blot analysis of IP samples treated with RNase A/T1 or EDTA after pulldown confirms the decrease in ribosomal proteins for both DCC and Nrp1

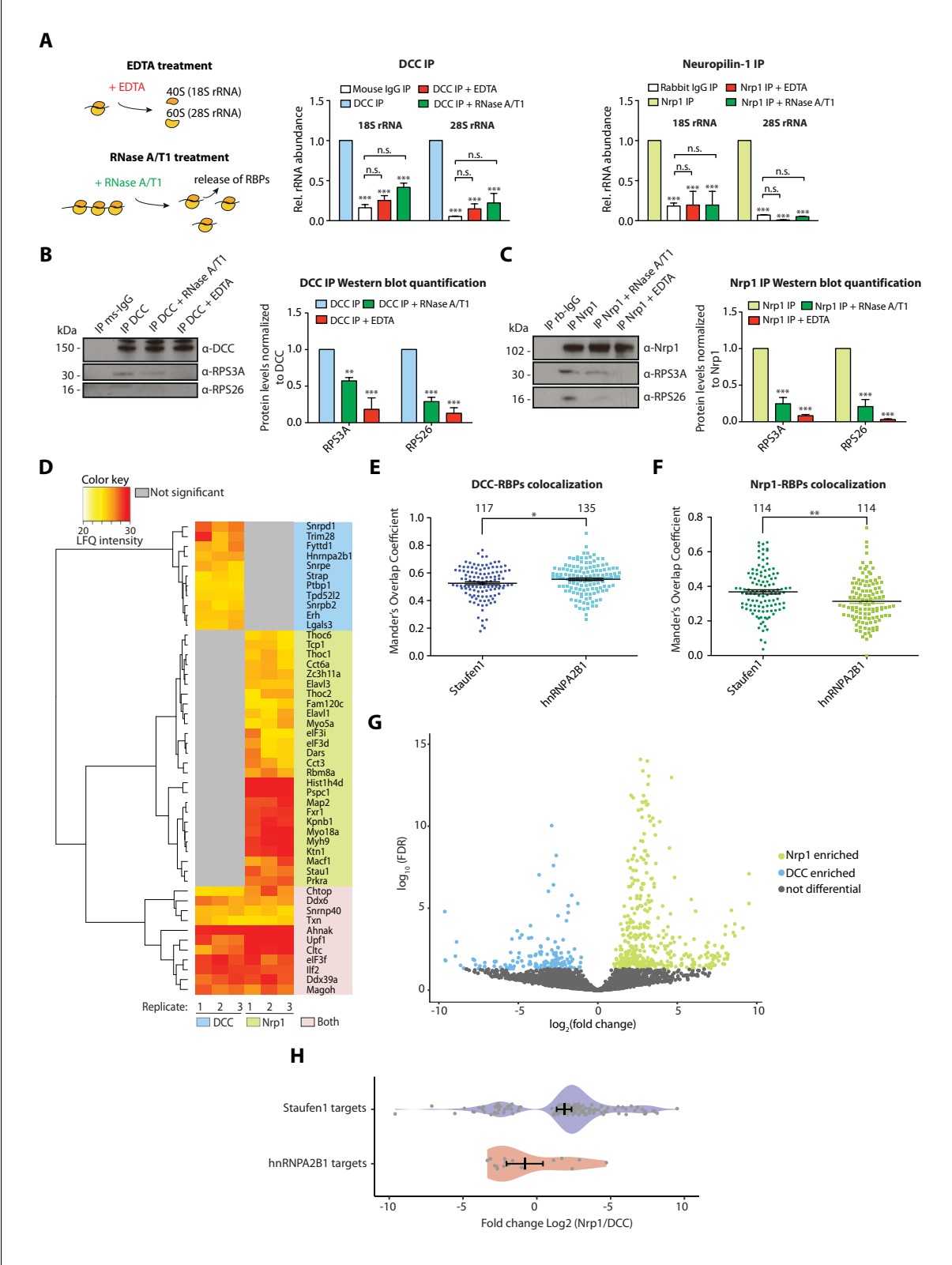

**Figure 2.** Receptor-ribosome coupling is mRNA dependent and DCC and Nrp1 bind to specific RBPs and mRNAs. (**A**) Relative 18S and 28S ribosomal RNA abundance after control (IgG) pulldown or receptors pulldowns with or without EDTA or RNase A/T1 treatments (two-way ANOVA with Bonferroni's multiple comparisons test; three biological replicates; Bars indicate mean, error bars indicate standard deviation; ***p<0.0001). (**B**) Western blot analysis and quantification of ribosomal proteins after DCC and (**C**) Nrp1 pulldowns. (two-way ANOVA with Bonferroni's multiple comparisons test;

*Figure 2 continued on next page*

*Figure 2 continued*

three biological replicates; Bars indicate mean, error bars indicate standard deviation; **p<0.01; ***p<0.0001). (D) Hierarchically-clustered heatmap of detected RBPs after DCC and Nrp1 pulldown. LFQ intensities are plotted for each IP-MS replicate. (E) Mander's overlap coefficients analysed using dual immunohistochemistry of DCC and Staufen1 or hnRNPA2B1 in axonal growth cones (unpaired two-tailed t-test; three biological replicates; individual data points are shown, error bars indicate SEM; p=0.03913). (F) Mander's overlap coefficients analysed using dual immunohistochemistry Nrp1 and Staufen1 or hnRNPA2B1 in axonal growth cones (unpaired two-tailed t-test; three biological replicates; individual data points are shown, error bars indicate SEM; p=0.00161). (G) Volcano plot showing differential expression analysis for DCC and Nrp1 pulldowns. (H) Enrichment analysis plot of known RBP targets of Staufen1 and hnRNPA2B1 detected in RNA-sequencing data after DCC and Nrp1 pulldown (individual data points are shown, error bars indicate standard deviation, Mann-Whitney test, Wilcoxon rank sum test DCC versus Nrp1; p=0.001511).

The online version of this article includes the following source data and figure supplement(s) for figure 2:

**Source data 1.** Spreadsheet containing all Manders Overlap Coefficient values for each axonal growth cone in *Figure 2E and F*.

**Source data 2.** Spreadsheet containing RNA-sequencing analysis of DCC and Nrp1 bound mRNAs and GO analysis of high abundant (FPKM >100) detected mRNAs for DCC and Nrp1.

**Figure supplement 1.** Polysome fractionation analysis, RNase sensitivity of Nrp1-Staufen1 interaction and additional RNA-seq analyses.

(*Figure 2B,C*), while the amounts of DCC and Nrp1 that precipitated were unaffected by the treatment conditions (*Figure 2B–C*). Together, these results suggest that the interaction of receptors with ribosomes is likely mediated through mRNA.

## DCC and Nrp1 bind to specific RNA-binding proteins

The mRNA-dependency of the receptor-ribosome interaction could be explained by mRNAs directly mediating the binding of receptors to ribosomes. Another possibility is that RNA binding proteins are key intermediaries in this binding and that mRNAs have a secondary role. Our MS analysis revealed that several RBPs are significantly enriched after DCC or Nrp1 pulldown (*Figure 2D*). Of 22 RBPs pulled down with DCC and 37 RBPs pulled down with Nrp1, only 11 are shared between the two receptors (*Figure 2D*). Several RBPs are significantly enriched in only one of the two receptor IPs. For example, Staufen1 is significantly enriched after Nrp1 IP, but not DCC IP, whereas hnRNPA2B1 is only detected after DCC IP (*Figure 2D*). This preferential RBP-receptor binding in axonal growth cones was also seen using dual immunocytochemistry with antibodies against DCC and Nrp1 and the RBPs Staufen1 and hnRNPA2B1 (*Figure 2E–F*). DCC co-localized with hnRNPA2B1 to a higher degree than with Staufen1 (*Figure 2E*). Conversely, Nrp1 showed a higher degree of co-localization with Staufen1 compared to hnRNPA2B1 (*Figure 2F*). RNAse A/T1 treatment was then used to test whether mRNA affects these associations. Western blot quantification after pulldown showed that the interaction of Staufen1 with Nrp1 was partly decreased by RNAse A/T1 treatment, suggesting that mRNA may stabilize the interaction between receptors and RBPs (*Figure 2—figure supplement 1E*). Together with our evidence implicating mRNA in the association of receptors with ribosomes, these results are consistent with a model in which receptors associate with specific RBPs, which bind specific mRNAs, and these mRNAs, in turn, recruit ribosomes.

## DCC and Nrp1 bind to specific subsets of mRNAs

Next, we examined if and which mRNAs can associate with DCC and Nrp1 by performing RNA-sequencing (RNA-seq) on RNAs isolated after DCC and Nrp1 IP. We used a human neuronal cell line, SH-SY5Y, for these experiments in order to rule out that any detected difference in the mRNAs is due to the expression of DCC and Nrp1 in different cell types. Co-precipitation of RNA was observed in DCC and Nrp1 pulldowns but not in IgG control pulldowns (*Figure 2—figure supplement 1F*). A distance matrix analysis revealed that the experimental replicates clustered together for each receptor and we observed a distinct signature of detected mRNAs between DCC, Nrp1 or whole lysate input samples (*Figure 2—figure supplement 1G*). Differential expression analysis revealed that DCC and Nrp1 each differentially bind to specific subsets of mRNAs, with 541 mRNAs differentially binding between DCC and Nrp1 (158 mRNAs for DCC *versus* 383 mRNAs for Nrp1) (*Figure 2G*). Of the highly abundant detected mRNAs (FPKM >100 and FPKM >1000), ~41% and ~70% respectively were differential between DCC and Nrp1, whilst with the low abundant detected mRNAs (FPKM 1–10), only ~5% were differential between DCC and Nrp1. GO enrichment analysis of both all and only high abundance (FPKM >100) differentially expressed mRNAs showed the receptor-specific enrichment of mRNAs involved in different processes (*Figure 2—figure*

*supplement 1H,I* and *Figure 2—source data 2*). For the high abundance mRNAs, GO terms that were associated with the mRNAs pulled down with DCC included 'cell-cell adhesion' and 'protein targeting', while 'translation' and 'small GTPase mediated signal transduction' were associated with Nrp1.

Although these results rely on mRNA populations expressed in SH-SY5Y cells, which may differ from mRNAs binding to these receptors in *Xenopus* RGC axons, we compared mRNAs that preferentially bind to DCC or Nrp1 (*Figure 2G*) with known mRNA targets of several RBPs (Staufen1, hnRNPA2B1, Elavl1 and Fxr1), which were identified by previous CLIP studies in other systems (*Lebedeva et al., 2011*; *Martinez et al., 2016*; *Sugimoto et al., 2015*; *Ascano et al., 2012*). In particular, we focused on Staufen1 and hnRNPA2/B1 because our proteomic analysis revealed that Staufen1 is enriched after Nrp1 pulldown compared to DCC pulldown and hnRNPA2B1 was only detected after DCC pulldown (*Figure 2D*). The analysis revealed significant enrichment of known targets of Staufen1 and hnRNPA2B1 in Nrp1 *versus* DCC pulldown, respectively (Mann-Whitney U test, Wilcoxon rank sum test; p=0.001511) (*Figure 2H*). Overall, the known targets of the 4 RBPs tested (Staufen1, hnRNPA2B1, Elavl1 and Fxr1) can account for 41.1% of the significantly enriched DCC-precipitated RNAs and for 43.1% of the significantly enriched Nrp1-precipitated mRNAs. Collectively, the results support a model where receptor-specific RBPs mediate the differential association of mRNAs to receptors.

## Receptor-ribosome coupling occurs in RGC axonal growth cones

As our IP experiments were performed in whole brain lysates (*Figure 1*), we next searched for evidence that these interactions occur in retinal growth cones. To begin to address this, we cultured eye primordia from *Xenopus* embryos and performed immunocytochemistry and expansion microscopy (*Chen et al., 2015*) on retinal axons using antibodies against the intracellular domain of DCC and a ribosomal protein (*Figure 3A*). DCC and RPL5/uL18 partially co-localized in retinal growth cones and filopodia (*Figure 3A*, white arrowheads). Similarly, RPS3A/eS1 co-localized with Nrp1 in retinal growth cones (*Figure 3B*, white arrowheads). Quantification of co-localization in expanded growth cones indicated a positive association between DCC and RPL5/uL18 (Pearson's correlation = 0.4316 ± 0.011, n = 73) and Nrp1 and RPS3A/eS1 (Pearson's correlation = 0.6727 ± 0.014, n = 72) (*Figure 3—figure supplement 1A*). To show close association of receptors and ribosomes in axonal growth cones, we employed the Proximity Ligation Assay (PLA) (*Söderberg et al., 2006*), modified for use on retinal axons (*Yoon et al., 2012*), which reports signal when the spatial coincidence of two proteins of interest is closer than 40 nm by using the respective antibodies. As a negative control, PLA was performed using the anti-DCC antibody and an IgG control antibody. This control generated a very low amount of background PLA signal (*Figure 3C*, *Figure 3—figure supplement 1B*), while we detected abundant PLA signal between DCC and RPL5/uL18, in line with previous findings (*Konopacki et al., 2016*), as well as with RPS4X/eS4 or RPL10A/uL1 (*Figure 3C*, *Figure 3—figure supplement 1B*). Similarly, Nrp1 generated abundant PLA signal together with RPS3A/eS1 or RPS23/uS12, with no detectable PLA signal in the negative control (Nrp1-IgG PLA) (*Figure 3D*). Given that EphB2 IP does not show any interaction with ribosomal proteins in *Xenopus* brain and eyes (*Figure 1F,J*), we tested whether this is conserved in retinal growth cones. Consistent with the IP results (*Figure 1F,J*) and with the EphB2-induced local protein synthesis independent growth cone collapse (*Mann et al., 2003*), PLA between EphB2 and RPL5/uL18 generated almost no detectable signal compared to DCC-RPL5/uL18 or Nrp1-RPS3A/eS1 in growth cones (*Figure 3E*). To provide further evidence, we performed electron microscopy on unstimulated axonal growth cones, and we observed a remarkable abundance of ribosomes in growth cones (*Figure 3F*). Strikingly, ribosomes could be seen aligned in rows underneath the plasma membrane (*Figure 3F*, *Figure 3—figure supplement 1C–E*), particularly in the regions in closest contact with the culture substrate. Indeed, we observed rows of ribosomes within 50 nm of the plasma membrane in 20 out of 22 axonal growth cones, and the presence of single 'isolated' ribosomes in the other two growth cones (*Figure 3F*, *Figure 3—figure supplement 1C*). The average distance between two neighboring ribosomes close to the plasma membrane in growth cones was significantly larger than the distance between ribosomes in the cell soma (58.12 ± 19.68 nm, n = 93 from 10 growth cones *versus* 23.05 ± 3.07 nm, n = 158 from five soma, p<0.00001) (*Figure 3G*, *Figure 3—figure supplement 1C,E*), indicative of and consistent with monosomes binding to the intracellular portions of transmembrane receptors, such as DCC or Nrp-1.

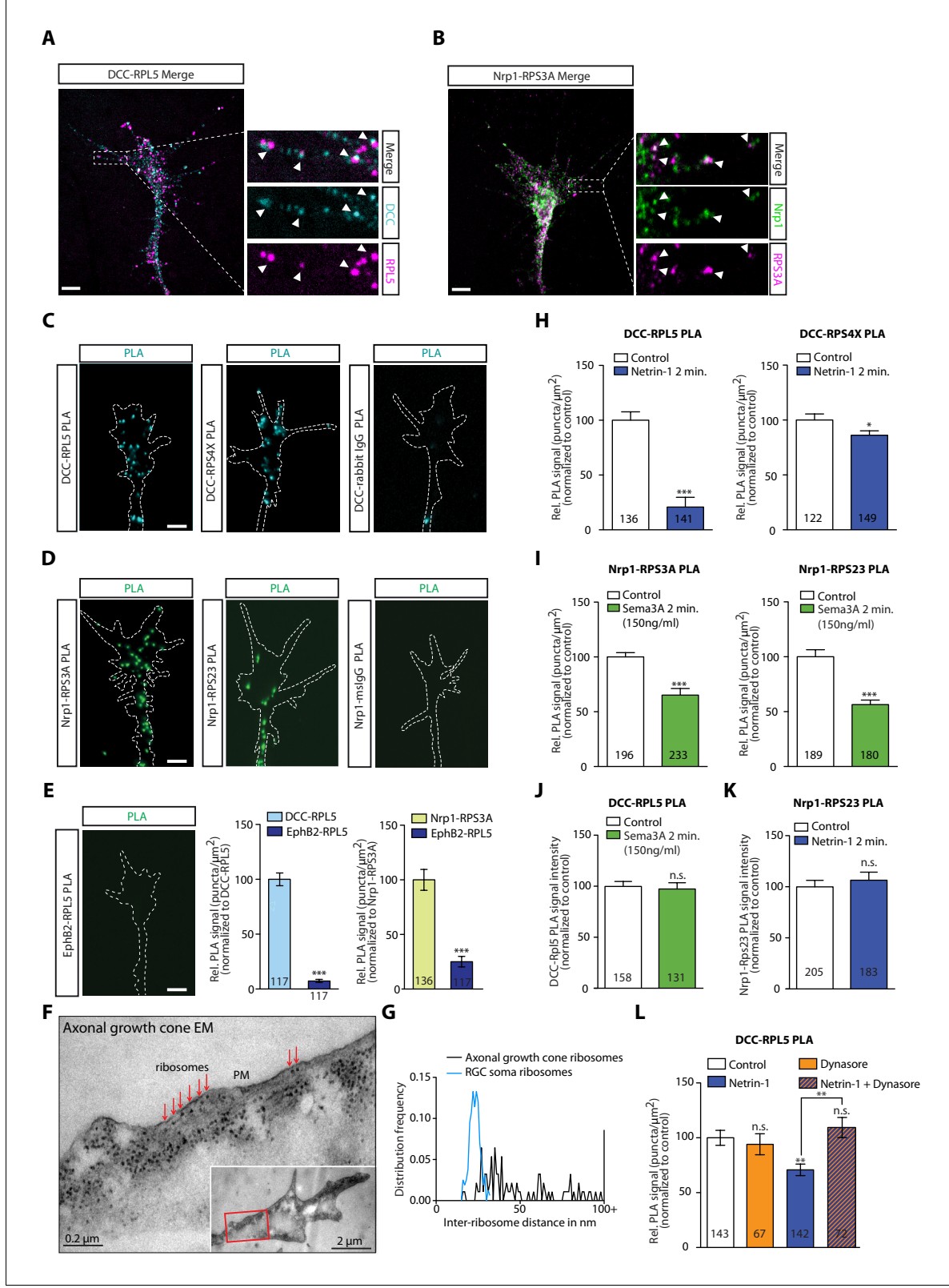

**Figure 3.** DCC and Nrp1 are in close proximity to ribosomes in axonal growth cones in a cue-dependent manner. (**A**) Expansion imaging shows partial co-localization of DCC and (**B**) Nrp1 with ribosomal proteins (Scale bars, 5 μm). (**C**) Representative proximity ligation assay signal in axonal growth cones between DCC and RPL5/uL18, RPS4X/eS4 or IgG control (Scale bars, 5 μm). (**D**) Representative proximity ligation assay signal in axonal growth cones between Nrp1 and RPS3A/eS1, RPS23/uS12 or IgG control (Scale bars, 5 μm). (**E**) Representative PLA signal in axonal growth cones between

*Figure 3 continued on next page*

*Figure 3 continued*

EphB2 and RPL5/uL18 (left) and quantification of PLA signal in axonal growth cones compared to DCC-RPL5/uL18 or Nrp1-RPS23/uS12 (right) (Mann-Whitney test; three biological replicates; bars indicate mean, error bars indicate SEM, ***p<0.0001; Scale bars, 5 μm). (F) EM image of an unstimulated axonal growth cone showing ribosomes aligned in a row (red arrows) under plasma membrane (PM). Inset shows the growth cone at lower magnification; the red box indicates the area shown in higher magnification. The section glances through the extreme surface of growth cone, where it attaches to the culture dish, giving rise to areas that lack subcellular structure. (G) Distribution frequency of the inter-ribosome distance in nm of ribosomes in axonal growth cones (n = 20) or in RGC soma (n = 5). All distances larger than 100 nm were pooled together. (H, I, J, K) Quantification of PLA signal in cue-stimulated axonal growth cones relative to control (unpaired two-tailed t-test; bars indicate mean, error bars indicate SEM; ***p<0.0001; *p=0.0423; for n.s. in J p=0.3522; for n.s. in K, p=0.885). (L) Relative PLA quantification of DCC and RPL5/uL18 compared to control after Dynasore pre-treatment (50 μM for 20 min), Netrin-1, or Netrin-1 + Dynasore (one-way ANOVA with Bonferroni's multiple comparisons test; bars indicate mean, error bars indicate SEM; p=0.001027 for Control vs. Netrin-1, p=0.000402 for Netrin-1 vs Netrin-1 + Dynasore, p=0.590377 for Control vs. Dynasore, p=0.384848 for Control vs Netrin + Dynasore). For all PLA experiments, numbers in bars indicate total number of growth cones quantified from at least three independent experiments.

The online version of this article includes the following source data and figure supplement(s) for figure 3:

**Source data 1.** Spreadsheet containing PLA counts and relative comparisons from each axonal growth cone in *Figure 3E*, all inter-ribosome distances and distribution shown in *Figure 3G*, and all normalized PLA count values for each axonal growth cone in *Figure 3H–L*.

**Figure supplement 1.** DCC and Nrp1 are in close proximity to ribosomes in axonal growth cones in a cue-dependent manner.

**Figure supplement 1—source data 1.** Spreadsheet containing all Pearson's correlation values for each expanded growth cone in *Figure 3—figure supplement 1A*, all normalized PLA count values for each axonal growth cone in *Figure 3—figure supplement 1F and H*, and all normalized puromycin intensity values for each axonal growth cone in *Figure 3—figure supplement 1G*.

## Dissociation of ribosomes from receptors is triggered by extrinsic cues and requires endocytosis

*Tcherkezian et al. (2010)* showed that ribosomes uncoupled from the DCC receptor in response to extracellularly applied Netrin-1, stimulating local translation, suggesting a mechanism for the precise spatiotemporal control of the proteome in subcellular compartments. Previous work has also shown that stimulation with the guidance cues Netrin-1 and Sema3A that bind DCC and Nrp1, respectively, triggers the remodelling of the axonal proteome within 5 min (*Cagnetta et al., 2018*). Therefore, we first asked whether the association between receptors and ribosomal proteins is cue-sensitive. Remarkably, the PLA signal between DCC and the ribosomal proteins RPL5/uL18 and RPS4X/eS4 decreased significantly in retinal axon growth cones after 2 min of Netrin-1 stimulation (*Figure 3H*), suggesting a rapid dissociation of ribosomes from the receptor. It should be noted that, whereas DCC protein level does not change in response to 5 min Netrin-1 stimulation, both RPL5/uL18 and RPS4X/eS4 are up-regulated in response to 5 min Netrin-1 stimulation (*Cagnetta et al., 2018*), indicating that the decrease in the PLA signal in response to Netrin-1 may be underestimated. In contrast to the DCC-RP PLA signal, the PLA signal between DCC and the RBP hnRNPA2B1 did not decrease after 2 min of Netrin-1 stimulation, indicating that the receptor-RBP interaction is not affected by cue stimulation (*Figure 3—figure supplement 1F*).

Extracellular Sema3A at a concentration of 150 ng/ml, which is known to affect local axonal translation (*Manns et al., 2012*; *Nédelec et al., 2012*), also triggers a significant decrease in the Nrp1-RPS3A/eS1 and RPS23/uS12 PLA signal within 2 min (*Figure 3I*). Interestingly, when Sema3A is presented extracellularly at a higher concentration (700 ng/ml), it induces growth cone collapse that is independent of protein synthesis (*Nédelec et al., 2012*; *Manns et al., 2012*). Puromycylation of newly synthesized proteins in axon-only cultures and subsequent visualization and quantification of immunofluorescence using an anti-puromycin antibody (*Schmidt et al., 2009*) in the presence of 700 ng/ml Sema3A shows no increase in global translation in growth cones (*Figure 3—figure supplement 1G*). In line with this finding, stimulation with 700 ng/ml Sema3A does not cause a rapid decrease in the Nrp1-RPS3A/eS1 PLA signal (*Figure 3—figure supplement 1H*). This suggested that the dissociation of ribosomes from Nrp1 in response to Sema3A is intimately linked to rapid and local protein synthesis. Importantly, the detected decrease in PLA signal is not due to changes in Nrp1, RPS3A/eS1 and RPS23/uS12 protein levels as these do not change in response to 5 min Sema3A stimulation (*Cagnetta et al., 2018*).

Next, we tested the specificity of the cue-induced dissociation of RPs from receptors by quantifying the PLA signal between DCC and RPL5/uL18 after Sema3A stimulation and the PLA signal between Nrp1 and RPS23/uS12 after Netrin-1 stimulation. In neither case did we observe a decrease

in PLA signal, confirming the ligand-receptor specificity of the cue-induced RP dissociation (*Figure 3J–K*).

The receptor-RP dissociation in response to an extrinsic cue suggests that this may occur on the plasma membrane but it is also possible that the dissociation happens intracellularly. Indeed, DCC and Nrp1 receptors are known to be rapidly endocytosed after cue stimulation (1–2 min) in growth cones (*Piper et al., 2005*) and we have recently identified the presence of ribosomal proteins on axonal endosomes which serve as platforms for local translation (*Cioni et al., 2019*), raising the possibility that the observed dissociation between receptors and ribosomes may also take place on endosomes. Therefore, we asked whether endocytosis plays a role in the cue-induced dissociation of ribosomes from receptors. Indeed, we found that treatment with the inhibitor of endocytosis Dynasore, a small GTPase inhibitor targeting dynamin (*Macia et al., 2006*), completely blocked the Netrin-1-induced decrease in PLA signal between DCC and RPL5/uL18, indicating that endocytosis is required for the receptor-ribosome dissociation (*Figure 3L*).

Together, these findings suggest that the rapid cue-specific dissociation of ribosomes in response to extracellular guidance cues is shared among different receptors, is tightly linked to cue-induced local translation-dependent responses, and requires endocytosis.

## Integration of multiple cues can affect the cue-induced selective translation of receptor-specific mRNAs

During axon pathfinding and branching, axons encounter multiple cues, such as EphrinB2 and Netrin-1,and can integrate these cues by forming a complex between their respective receptors in a ligand-dependent manner (*Morales and Kania, 2017*; *Dudanova and Klein, 2013*; *Poliak et al., 2015*). The cue EphrinA1 has been reported to decrease local translation in hippocampal axons (*Nie et al., 2010*) and the rapid local translation of the Translationally controlled tumor protein (Tctp), which is up-regulated by Netrin-1 (*Gouveia Roque and Holt, 2018*). Therefore, we asked whether co-stimulation with EphrinA1 and Netrin-1 alters the dissociation of ribosomes from DCC. To address this question, we co-stimulated retinal axons with Netrin-1 and EphrinA1 and examined receptor-ribosome coupling using the PLA approach. Whereas Netrin-1 induces a decrease in the DCC-RPL5/uL18 PLA signal within 2 min, both Ephrin-A1 stimulation alone and co-stimulation with Netrin-1 and EphrinA1 do not decrease the DCC-RPL5/uL18 PLA signal, indicating that the Netrin-1-induced dissociation of ribosomes from DCC is blocked by co-stimulation with EphrinA1 (*Figure 4A*). By contrast, co-stimulation with EphrinA1 and Sema3A does not block the Sema3A-induced decrease in the Nrp1-RPS23/uS12 PLA signal (*Figure 4—figure supplement 1A*). These results reveal that integration of guidance cues can alter the receptor-ribosome dissociation, possibly by structural changes of the interacting receptors (*Morales and Kania, 2017*; *Dudanova and Klein, 2013*; *Poliak et al., 2015*).

Our data showing that EphrinA1 blocks the Netrin-1-induced ribosome dissociation from DCC, suggest that EphrinA1 may inhibit the axonal translation induced by Netrin-1. To test this hypothesis, we examined the effect of cue integration of Netrin-1 and EphrinA1 on both global and selective local translation in growth cones. In the culture conditions used in this study (*Höpker et al., 1999*), both Netrin-1 and EphrinA1 decrease global local translation in axons as measured by the puromycylation assay in axon-only cultures (*Figure 4B–C*). Consistent with this result, both cues decrease pERK1/2 levels (*Figure 4—figure supplement 1B*), an upstream activator of the TOR signalling pathway, which is known to regulate axonal protein synthesis (*Campbell and Holt, 2003*).

Despite the decrease in global axonal translation, previous work has revealed that Netrin-1 can induce the rapid selective translation of specific mRNAs (*Cagnetta et al., 2018*; *Shigeoka et al., 2019*). The IP-RNA-seq data in human SH-SY5Y cells had revealed that DCC associates with mRNAs encoding β-catenin (*ctnnb1*) and hnRNPH1 (*hnrnph1*) significantly more than with Nrp1. Interestingly, *ctnnb1* and *hnrnph1* mRNAs have been detected in *Xenopus* retinal axons (*Shigeoka et al., 2019*) and are selectively synthesised in response to 5 min Netrin-1 stimulation, but not Sema3A (*Cagnetta et al., 2018*), indicating that receptor-specific mRNAs can underlie the cue-induced selective translation. To further test this, we examined whether these mRNAs associate with DCC also in *Xenopus* brain and eyes by carrying out IP followed by qPCR. The results showed significant enrichment of *ctnnb1* and *hnrnph1* mRNAs in DCC pulldown compared to an IgG pulldown, thus confirming their association with DCC (*Figure 4D*). Finally, quantification of immunofluorescence confirmed that both β-catenin and hnRNPH1 protein levels increase in response to 5 min Netrin-1 stimulation,

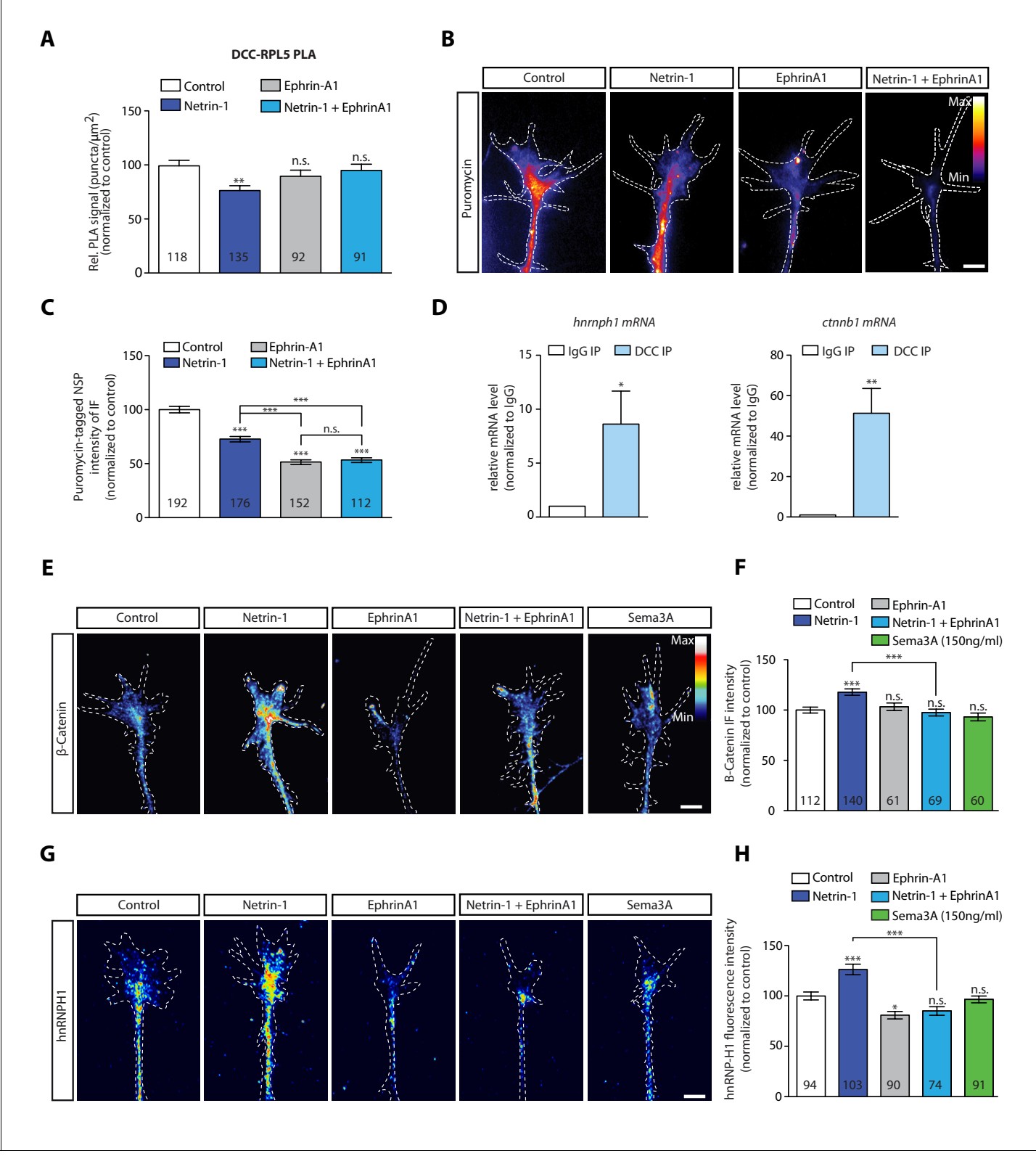

**Figure 4.** EphrinA1 co-stimulation blocks Netrin-1 induced receptor-ribosome dissociation and selective translation. (**A**) Relative PLA quantification of DCC and RPL5/uL18 compared to control after Netrin-1, EphrinA1, or co-stimulation (one-way ANOVA with Bonferroni's multiple comparisons test; bars indicate mean, error bars indicate SEM; **p<0.01). (**B, C**) Puromycin QIF relative to control after Netrin-1, EphrinA1 or co-stimulation (one-way ANOVA with Bonferroni's multiple comparisons test; bars indicate mean, error bars indicate SEM; ***p<0.0001). (**D**) Relative mRNA quantification after DCC IP

*Figure 4 continued on next page*

*Figure 4 continued*

of *hnrnph1* and *ctnnb1* mRNA (unpaired t-test with Welch's corrections on dCT values; three biological replicates; bars indicate mean, error bars indicate SEM; *p=0.02 for *hnrnph1*; **p=0.0018 for *ctnnb1*). (E, F) β-Catenin QIF relative to control after Netrin-1, EphrinA1, Sema3A or Netrin-1 and EphrinA1 co-stimulation (one-way ANOVA with Bonferroni's multiple comparisons test; bars indicate mean, error bars indicate SEM; ***p<0.0001). (G, H) hnRNPH1 QIF relative to control after Netrin-1, EphrinA1, Sema3A or Netrin-1 and EphrinA1 co-stimulation (one-way ANOVA with Bonferroni's multiple comparisons test; bars indicate mean, error bars indicate SEM; ***p<0.0001; *p=0.0164). Scale bars, 5 µm. For all QIF experiments, numbers in bars indicate amount of growth cones quantified collected from at least three independent experiments.

The online version of this article includes the following source data and figure supplement(s) for figure 4:

**Source data 1.** Spreadsheet containing all normalized PLA count values for each axonal growth cone in *Figure 4A*, all normalized puromycin intensity values for each axonal growth cone in *Figure 4C*, all normalized ß-Catenin intensity values for each axonal growth cone in *Figure 4F* and all normalized hnRNPH1 intensity values for each axonal growth cone in *Figure 4H*.

**Figure supplement 1.** EphrinA1 co-stimulation blocks Netrin-1 induced receptor-ribosome dissociation and selective translation of *rps14*.

**Figure supplement 1—source data 1.** Spreadsheet containing all normalized PLA count values for each axonal growth cone in *Figure 4—figure supplement 1A*, all normalized pERK1/2 intensity values for each axonal growth cone in *Figure 4—figure supplement 1B* and all normalized RPS14 intensity values for each axonal growth cone in *Figure 4—figure supplement 1D and E*.

but not Sema3A (*Figure 4E–H*), in line with previous axonal translation findings (*Cagnetta et al., 2018*).

Similar to β-catenin and hnRNPH1, RPS14/uS11 mRNA is present in *Xenopus* retinal axons (*Shigeoka et al., 2019*) and is up-regulated in response to 5 min Netrin-1 stimulation, but not Sema3A (*Cagnetta et al., 2018*), as confirmed by quantification of immunofluorescence (*Figure 4—figure supplement 1E*). However, *rps14* mRNA was not detected to be associated with DCC in SH-SY5Y cells. Therefore, we asked whether this is due to interspecies differences (human (SH-SY5Y) *versus Xenopus*), or whether *rps14* is selectively translated via a DCC interactome-independent mechanism. To address this question, we carried out IP followed by qPCR in *Xenopus* brain and eyes, which confirmed *rps14* association to DCC (*Figure 4—figure supplement 1C*). Our findings that Netrin-1, but not Sema3A, induces the translation of mRNAs bound to DCC point towards a model where receptor-specific mRNA interactomes act as a hub for rapid cue-specific selective translation.

Finally, we examined the effect of EphrinA1 co-stimulation on the Netrin-1-induced selective translation up-regulation of β-catenin, hnRNPH1 and RPS14/uS11. Quantification of immunofluorescence showed that EphrinA1 stimulation alone does not affect β-catenin and RPS14/uS11 protein levels (*Figure 4E–H*; *Figure 4—figure supplement 1D*) and decreases hnRNPH1 protein level in axonal growth cones (*Figure 4G–H*). Co-stimulation with Netrin-1 and EphrinA1 blocks the Netrin-1-induced increase of all three proteins (*Figure 4E–H*; *Figure 4—figure supplement 1D*). Together, the results show that integration of the EphrinA1 and Netrin-1 signals inhibits the Netrin-1-induced selective translation, possibly by inhibiting DCC-ribosome dissociation (*Figure 4A*).

## Discussion

We provide evidence for a receptor-ribosome coupled mechanism by which extrinsic cues cause rapid and selective changes in the local proteome. In support of this model, we show that multiple guidance cue receptors interact with ribosomes, that the interaction between receptors and ribosomes depends on mRNA and rapidly decreases within 2 min of cue stimulation. Moreover, we find that receptors bind to distinct subsets of RBPs and mRNAs, and that cue stimulation induces the selective axonal translation of several receptor-specific mRNAs. Finally, we show that the integration of multiple cues can alter receptor-ribosome dissociation and selective translation.

Based on the candidate receptors tested here, we suggest that whether or not a particular receptor shows receptor-ribosome coupling is related to whether or not the receptors regulate local translation upon ligand binding. Future studies are needed to determine whether receptor-ribosome coupling is restricted to axon guidance receptors and neurons. Interestingly, a previous study has reported the association of a chemokine receptor, CXCR4, with eukaryotic initiation factor 2B (eIF2B), which decreases upon ligand binding in a pre-B cell line (*Palmesino et al., 2016*). In addition, several adrenergic receptor subtypes have been reported to associate with eIF2B at the plasma membrane (*Klein et al., 1997*). This raises the intriguing possibility that coupling of translational

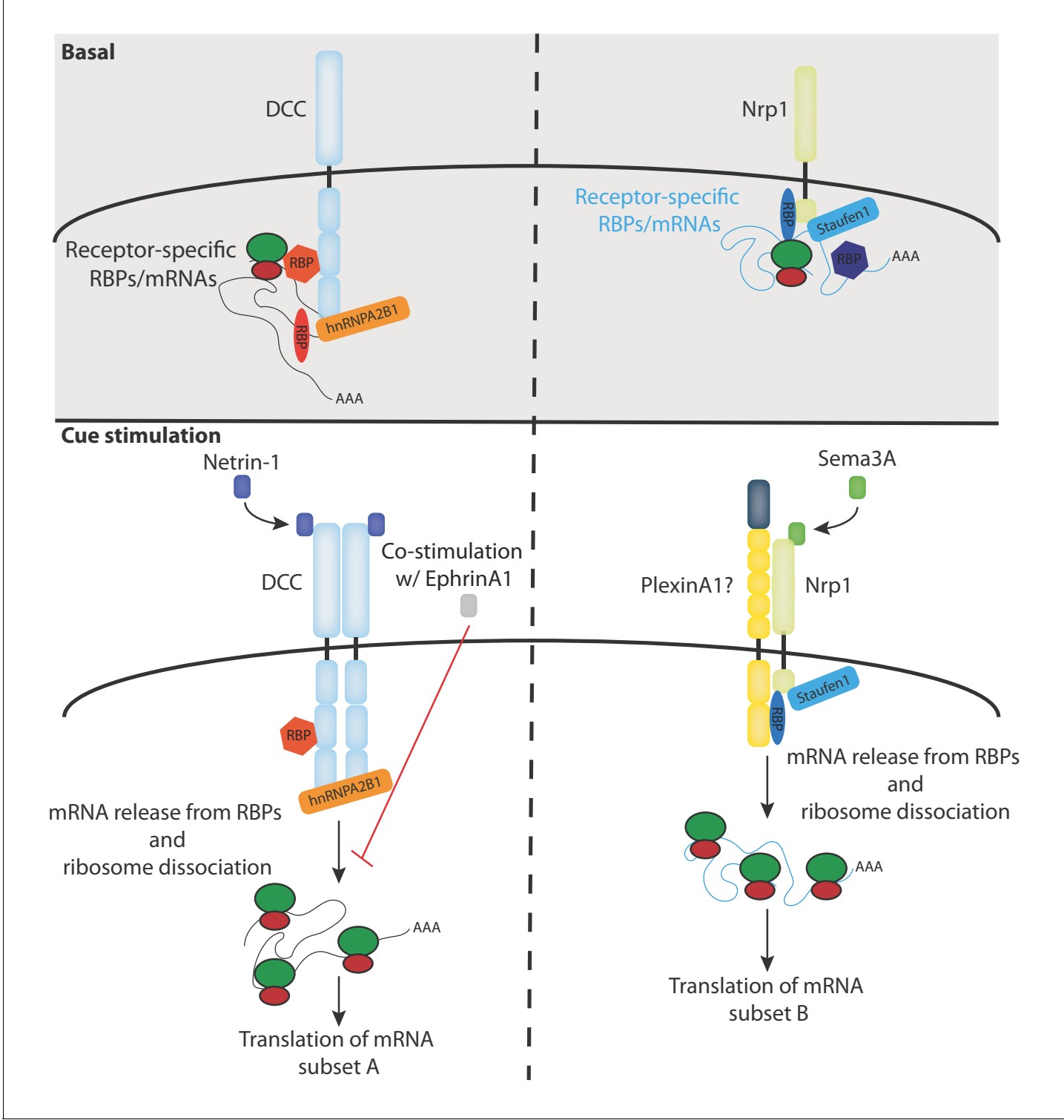

**Figure 5.** Model diagram depicting the proposed interactions between receptors, RBPs, mRNAs and ribosomes under basal and cue stimulation conditions.

machinery with receptors extends to other cell types and is a widespread mechanism to rapidly transduce local translation downstream of extracellular signals.

Previous studies have shown that the RBP zipcode binding protein 1 (ZBP1) can be phosphorylated upon cue stimulation, thereby regulating local translation in axons by possibly releasing the bound mRNAs (*Hüttelmaier et al., 2005*; *Sasaki et al., 2010*; *Lepelletier et al., 2017*). DCC and Nrp1 each differentially bind to RBPs and mRNAs, thus providing a way to rapidly achieve cue-induced selective translation. We observed an enrichment of known mRNA targets for RBPs detected specifically in DCC and Nrp1 pulldowns respectively, suggesting a role for RBPs in mediating the differential binding of mRNAs to receptors and their cue-induced selective translation. This hypothesis is supported by the enrichment of the RBP hnRNPA2B1 and *ctnnb1* mRNA (encoding β-catenin) specifically in DCC but not Nrp1 pulldown, as hnRNPA2B1 has been reported to control the translation of β-catenin (*Stockley et al., 2014*), which is selectively translated in response to Netrin-1, but not Sema3A in retinal axons (*Cagnetta et al., 2018*), in accord with the data reported here.

Our RNA-seq analysis reveals a receptor-specific enrichment of 100–400 mRNAs suggesting that a large number of mRNAs may be regulated by specific receptors and their ligands (*Figure 2G*). This idea is consistent with our previous proteomics study in *Xenopus* retinal axons showing that the translation of more than 100 mRNAs is regulated within 5 min in response to Netrin-1 and Sema3A (*Cagnetta et al., 2018*). It should be noted that, as our RNA-seq data are obtained from the human cell line SH-SY5Y, the number, and exact identity, of receptor-associated mRNAs may be different in axons. This is exemplified by the absence of *rps14* mRNA enrichment in SH-SY5Y cells, which was detected in *Xenopus* brains (*Figure 4—figure supplement 1C*). In addition, it is possible that not all detected mRNAs interact with DCC and Nrp1 at the plasma membrane as a portion of these mRNAs could also be associated with receptors on endocytic vesicles that are known to contain DCC and Nrp1. Our results point to a model in which different subsets of mRNAs interact via specific RBPs with either DCC or Nrp1, and are released, together with ribosomes, upon specific cue stimulation and thus become available for subsequent translation (*Figure 5*). To fully understand and validate our model, it will be key to investigate the complex inter-dependency of these interactions.

It should be noted that, in addition to RBPs and mRNAs, several other molecules characterize the receptor-specific interactome. For example, eIF3d, an initiation factor previously shown to regulate specialized translation initiation, is significantly enriched specifically after Nrp1 IP, but not DCC IP, thus raising the interesting possibility that differential binding to initiation factors may contribute to cue-induced selective translation (*Lee et al., 2016*). Intriguingly, a recent study revealed that an untranslated mRNA can associate with and regulate the signalling of the TrkA receptor in axons via its axon-enriched long 3'UTR (*Crerar et al., 2019*). It will be interesting to investigate whether any of the DCC and Nrp1 targets identified in our study also play a structural role, for example by regulating the receptor-ribosome association and/or the downstream signalling and local translation.

During axon guidance and branching, axons can encounter a combination of extracellular signals and ample evidence shows that the integration of multiple cues results in different outcomes than those of each single cue (*Dudanova and Klein, 2013*; *Morales and Kania, 2017*). Here, we tested the effect of cue integration on receptor-ribosome coupling and found that EphrinA1 blocks the Netrin-1-induced ribosome dissociation from DCC, but not the Sema3A-induced ribosome dissociation from Nrp1. In addition, EphrinA1 blocks the Netrin-1-induced selective increase in translation of several mRNAs. The mechanism by which EphrinA1 affects the coupling of DCC to ribosomes is unknown. One possibility is that, upon co-stimulation of EphrinA1 and Netrin-1, the DCC and Eph receptors may form a complex, thereby altering the receptor structure and association to ribosomes, which could be consistent with a previous study revealing a ligand-dependent interaction between the receptors Unc5 and EphB2 (*Poliak et al., 2015*).

In conclusion, our findings show that coupling of the translational machinery to guidance cue receptors at the plasma membrane of growth cones is a mechanism to rapidly and selectively control the cue-induced regulation of the local proteome and suggest that this may be a general principle that applies to membrane receptors more broadly.

# Materials and methods

## Key resources table

| Reagent type (species) or resource | Designation | Source or reference | Identifiers | Additional information |
|---|---|---|---|---|
| Biological sample (*Xenopus laevis*) | *Xenopus laevis* | NASCO | Cat# LM00715 (male); RRID:XEP_Xla100; Cat# LM00535 (female); RRID:XEP_Xla | |
| Cell line (*Homo-sapiens*) | SH-SY5Y | ATCC | Cat# CRL-2266; RRID:CVCL_0019 | |
| Antibody | anti-RPS3A (Rabbit polyclonal) | Abcam | Cat# ab194670; RRID:AB_2756396 | ICC/PLA (1:100) WB (1:1000) |
| Antibody | Anti-Neuropilin-1 (Rabbit monoclonal) | Abcam | Cat# ab81321; RRID:AB_1640739 | ICC/PLA (1:100) WB (1:2000) IP (5 μg) |
| Antibody | Anti-Neuropilin-1 (Mouse monoclonal) | Proteintech | Cat# 60067–1-Ig; RRID:AB_2150840 | ICC (1:100) |
| Antibody | Anti-DCC (mouse monoclonal | BD Biosciences | Cat# 554223; RRID:AB_395314 | ICC/PLA (1:100) WB (1:1000) IP (5 μg) |
| Antibody | Anti-RPL5 (rabbit polyclonal | Proteintech | Cat# 15430–1-AP; RRID:AB_2238681 | ICC/PLA (1:100) |
| Antibody | Anti-RPS4X (Rabbit polyclonal) | Proteintech | Cat# 14799–1-AP; RRID:AB_2238567 | PLA (1:100) WB (1:1000) |
| Antibody | Anti-RPL10A (Rabbit polyclonal) | Proteintech | Cat# 16681–1-AP; RRID:AB_2181281 | PLA (1:100) WB (1:500) |
| Antibody | Anti-RPS23 (mouse monoclonal) | Abcam | Cat#: ab57644; RRID:AB_945314 | PLA (1:100) WB (1:1000) |
| Antibody | Anti-RPS26 (Rabbit polyclonal | Proteintech | Cat# 14909–1-AP; RRID:AB_2180361 | WB (1:500) |
| Antibody | Anti-Robo2 (goat polyclonal) | R and D Systems | Cat# AF3147; RRID:AB_2181857 | WB (1:250) |
| Antibody | Anti-EphB2 (mouse monoclonal) | Santa Cruz | Cat# sc130068; RRID:AB_2099958 | WB (1:100) IP (5 μg) |
| Antibody | Anti-EphB2 (mouse monoclonal) | Thermo Fisher Scientific | Cat# 37–1700; RRID:AB_2533302 | PLA (1:100) |
| Antibody | Anti-Staufen1 (Rabbit polyclonal) | Abcam | Cat# ab73478; RRID:AB_1641030 | ICC (1:100) WB (1:500) |
| Antibody | Anti-hnRNPA2B1 (Rabbit polyclonal) | Abcam | Cat# ab31645; RRID:AB_732978 | ICC/PLA (1:100) |
| Antibody | Anti-RPS14 (Rabbit polyclonal) | Abcam | Cat# ab174661 | ICC (1:100) |
| Antibody | Anti-ß-Catenin (Rabbit polyclonal) | Sigma-Aldrich | Cat# C2206; RRID:AB_476831 | ICC (1:500) |
| Antibody | Anti-hnRNPH1 | Abcam | Cat# ab154894 | ICC (1:500) |
| Antibody | Anti-IgG (Rabbit) | Abcam | Cat# ab37415; RRID:AB_2631996 | PLA (1:100) IP (5 μg) |
| Antibody | Anti-IgG1 (Mouse) | R and D Systems | Cat# MAB002; RRID:AB_357344 | PLA (1:100) IP (5 μg) |
| Antibody | Anti-IgG2b (Mouse) | R and D Systems | Cat# MAB004; RRID:AB_357346 | IP (5 μg) |
| Antibody | Anti-IgG (Goat) | R and D Systems | Cat# AB-108-C; RRID:AB_354267 | IP (5 μg) |

*Continued on next page*

Continued

| Reagent type (species) or resource | Designation | Source or reference | Identifiers | Additional information |
|---|---|---|---|---|
| Antibody | Anti-Puromycin-Alexa Fluor 488 conjugate (mouse monoclonal) | Millipore | Cat# MABE343-AF488; RRID:AB_2736875 | ICC (1:200) |
| Antibody | Anti-RPL19 (mouse monoclonal) | Abcam | Cat#ab58328; RRID:AB_945305 | WB (1:1000) |
| Antibody | Anti-FxR | Gift from Dr. Edward Khandjan, University of Quebec | N/A | WB (1:1000) |
| Antibody | Anti-pERK1/2 | Cell Signaling | Cat# 9101; RRID:AB_331646 | ICC (1:250) |
| Antibody | Goat-anti-rabbit Alexa Fluor 568 | Abcam | Cat# ab150077; RRID:AB_2630356 | ICC (1:1000) |
| Antibody | Goat-anti-mouse Alexa Fluor 568 | Abcam | Cat# ab150117; RRID:AB_2688012 | ICC (1:1000) |
| Antibody | Goat-anti-mouse-HRP | Abcam | Cat# ab6789; RRID:AB_955439 | WB (1:15000) |
| Antibody | Goat-anti-rabbit-HRP | Abcam | Cat#: ab97080; RRID:AB_10679808 | WB (1:15000) |
| Commercial assay or kit | RNeasy mini kit | Qiagen | Cat# 74104 | |
| Commercial assay or kit | SuperScript III First-strand Synthesis kit | Thermo Fisher Scientific | Cat# 18080051 | |
| Commercial assay or kit | Quantitect SYBR green PCR kit | Qiagen | Cat# 204143 | |
| Commercial assay or kit | KAPA HyperPrep kit | Roche | Cat# KK8503 | |
| Commercial assay or kit | NextSeq 500/550 high output v2 kit (150 cycles) | Illumina | Cat# FC-404–2002 | |
| Commercial assay or kit | Duolink In situ PLA Detection reagents green | Sigma-Aldrich | Cat# DUO92014 | |
| Commercial assay or kit | Duolink In situ PLA Detection reagents red | Sigma-Aldrich | Cat# DUO92008 | |
| Commercial assay or kit | Duolink In situ PLA probe Anti-Rabbit PLUS | Sigma-Aldrich | Cat# DUO92002 | |
| Commercial assay or kit | Duolink In situ PLA probe Anti-Mouse MINUS | Sigma-Aldrich | Cat# DUO92004 | |
| Chemical compound, drug, reagent | Cycloheximide | Sigma Aldrich | Cat# C4859 | |
| Chemical compound, drug, reagent | RNase A | Ambion | Cat# EN0531 | |
| Chemical compound, drug, reagent | RNase T1 | Ambion | Cat# EN0541 | |

*Continued*

| Reagent type (species) or resource | Designation | Source or reference | Identifiers | Additional information |
|---|---|---|---|---|
| Chemical compound, drug, reagent | Puromycin | Sigma-Aldrich | Cat# P8833 | |
| Chemical compound, drug, reagent | Recombinant mouse Netrin-1 | R and D systems | Cat# 1109-N1 | |
| Chemical compound, drug, reagent | Recombinant human Sema3A | R and D systems | Cat# 1250-S3 | |
| Chemical compound, drug, reagent | Dynasore | Sigma-Aldrich | Cat# D7693 | |
| Chemical compound, drug, reagent | SUPERase In RNAse inhibitor | Ambion | Cat# AM2696 | |
| Software, algorithm | Volocity | PerkinElmer | Version 6.0.1; RRID:SCR_002668 | |
| Software, algorithm | GraphPad Prism | GraphPad | v.5; RRID:SCR_002798 | |
| Software, algorithm | R | Other | v.3.2.2; RRID:SCR_001905 | https://www.r-project.org |
| Software, algorithm | MATLAB | Mathworks | v.R2016b; RRID:SCR_001622 | |
| Software, algorithm | HISAT2 | Other | v.2.1.0; RRID:SCR_015530 | https://ccb.jhu.edu/software/hisat2/index.shtml |
| Software, algorithm | Cufflinks | Other | v.2.2.1; RRID:SCR014597 | http://cole-trapnell-lab.github.io/cufflinks/ |

## Embryos

*Xenopus laevis* embryos were fertilized in vitro and raised in 0.1x Modified Barth's Saline (8.8 mM NaCl, 0.1 mM KCl, 0.24 mM NaHCO$_3$, 0.1 mM HEPES, 82 µM MgSO$_4$, 33 µM Ca(NO$_3$)$_2$, 41 µM CaCl$_2$) at 14–20°C and staged according to the tables of *Nieuwkoop and Faber (1994)*. All animal experiments were approved by the University of Cambridge Ethical Review Committee in compliance with the University of Cambridge Animal Welfare Policy. This research has been regulated under the Animals (Scientific Procedures) Act 1986 Amendment Regulations 2012 following ethical review by the University of Cambridge Animal Welfare and Ethical Review Body (AWERB). All animals used in this study were below stage 45.

## Cell line culture

Human neuroblastoma SH-SY5Y cells (ATCC; Cat# CRL-2266), free of mycoplasma, were cultured in Dulbecco's minimal essential medium (DMEM) containing antibiotics, L-glutamine and 10% fetal bovine serum (FBS).

## Primary *Xenopus* retinal cultures

Eye primordia were dissected from Tricaine Methanesulfonate (MS222) (Sigma-Aldrich) anesthetized embryos at stage 35/36 (or stage 32 for EM) and cultured on 10 µg/ml poly-L-lysine- (Sigma-Aldrich) and 10 µg/ml laminin- (Sigma-Aldrich) coated dishes in 60% L-15 medium (Gibco) at 20°C for 24 hr before performing immunohistochemistry or proximity ligation assay, or for 48 hr before the puromycilation assay. Where indicated in the figures and figure legends, cultures were treated with Netrin-1 (600 ng/ml, R and D systems, 1109-N1), Sema3A (150 or 700 ng/ml, R and D systems, 1250-S3), or Dynasore (50 µM, Sigma-Aldrich, D7693).

## Immunoprecipitation

SH-SY5Y cells or *Xenopus* brains and eyes dissected from stage 40/41 embryos were lysed in lysis buffer (20 mM Tris-HCl, pH 7.4, 150 mM NaCl, 10 mM MgCl2 and 10% glycerol supplemented with 100 µg/ml cycloheximide (Sigma-Aldrich), EDTA-free protease inhibitors (Roche, 11873580001), phosphatase inhibitors (Thermo Fisher Scientific, A32957) and SuperRNAse In RNAse inhibitor (Ambion, AM2696)). Tissues or cells were lysed for 30 min at 4˚C and centrifuged for 5 min at 800 g at 4˚C to remove unlysed cells and nuclei and then 15 min at 16000 g at 4˚C. The resulting supernatant was incubated with magnetic Dynabeads pre-coupled with antibodies using the Dynabeads antibody coupling kit (Thermo Fisher Scientific, 14311D) for 1.5 hr at 4˚C on a rotor. The following antibodies were used: mouse-anti-DCC (BD Biosciences, 554223); rabbit-anti-Nrp1 (Abcam, ab81321); goat-anti-Robo2 (R and D systems, AF3147); mouse-anti-EphB2 (Santa Cruz, sc130068) or an isotype control: rabbit IgG (Abcam, ab37415); mouse IgG1 (R and D systems, MAB002); mouse IgG2b (R and D systems, MAB004); goat IgG (R and D systems, AB-108-C). Beads were then washed three times in lysis buffer and processed for protein or RNA isolation. For EDTA and RNase A/T1 treatment pulldowns, immunoprecipitated samples (samples after incubation of supernatant with antibody-coupled beads) were equally divided into three tubes (tube 1: normal washes as above, tube 2: EDTA treatment washes, tube 3: RNase A/T1 treatment washes). For EDTA treatment, immunoprecipitated samples were washed with EDTA wash buffer (20 mM Tris-HCl, pH 7.4, 150 mM NaCl, 25 mM EDTA and 10% glycerol supplemented with EDTA-free protease inhibitors (Roche, 11873580001), phosphatase inhibitors (Thermo Fisher Scientific, A32957) for three times before elution. For RNase A/T1 treatment, immunoprecipitated samples were washed three times for 3 min at RT with RNase A/T1 wash buffer (20 mM Tris-HCl, pH 7.4, 150 mM NaCl, 10 mM MgCl2% and 10% glycerol supplemented with 100 µg/ml cycloheximide (Sigma-Aldrich), EDTA-free protease inhibitors (Roche, 11873580001), phosphatase inhibitors (Thermo Fisher Scientific, A32957), 10 µg/µl RNase A (Ambion, EN0531) and 250U RNase T1 (Ambion, EN0541). After normal, EDTA, or RNase A/T1 washes, samples were processed for protein or RNA isolation.

For protein isolation, 1x NuPAGE LDS sample buffer (Thermo Fisher Scientific, NP0008) was added to the beads, incubated for 5 min at 95˚C and the final protein eluate was collected after magnetic separation of the beads. For RNA isolation, RLT buffer was added to the beads, vortexed for 2 min and then separated from the beads on a magnetic stand.

## Polysome fractionation

For density gradient fractionation, lysate was layered on a sucrose gradient (10–50%) in PLB buffer (20 mM Tris-HCl, pH 7.4, 150 mM NaCl, 10 mM MgCl2, 100 µg/ml cycloheximide (Sigma-Aldrich), 0.5 mM DTT) and ultracentrifugation was performed using a Beckman SW-40Ti rotor and Beckman Optima L-100 XP ultracentrifuge, with a speed of 35,000 rpm at 4˚C for 160 min. Fractionations and UV absorbance profiling were carried out using Density Gradient Fractionation System (Teledyne ISCO). Proteins were precipitated from each fraction using methanol-chloroform precipitation and pellets were resuspended in 1x NuPAGE LDS sample buffer and used for Western blotting as described below.

## Western blot

Proteins were resolved by SDS-PAGE on NuPage 4–12% Bis-Tris gels (Invitrogen, NP0321) and transferred to nitrocellulose membrane (Bio-Rad). The blots were blocked in 5% milk in TBST-T for 60 min at RT and then incubated with primary antibodies in 5% milk in TBS-T overnight at 4˚C. After washing three times with TBS-T the blots were incubated with HRP-conjugated secondary antibodies (goat-anti-mouse HRP (Abcam, ab6789); goat-anti-rabbit HRP (Abcam, ab6721) for 1 hr at RT, washed again for three times in TBS-T, followed by ECL-based detection (Pierce ECL plus, Thermo Scientific, 32123). The following primary antibodies were used for Western blot analysis: mouse-anti-DCC (BD Biosciences, 554223), rabbit-anti-neuropilin-1 (Abcam, ab81321), goat-anti-Robo2 (R and D systems, AF3147), mouse-anti-EphB2 (Santa Cruz, sc130068), mouse anti-Rpl19/eL19 (Abcam, ab58328), mouse anti-RPS23/uS12 (Abcam, ab57644), rabbit anti-RPS4X/eS4 (Proteintech, 14799–1-AP), rabbit-anti RPL10A/uL1 (Proteintech, 16681–1-AP), rabbit-anti Rps26 (Proteintech, 14909–1-AP), mouse-anti-Rps3A (Abcam, ab194670), mouse-anti-FxR (gift from dr. Khandjian), rabbit-anti-Staufen1 (Abcam, ab73478).

## Quantitative RT-PCR

RNA was isolated from eluted samples using the RNeasy Mini kit (Qiagen, 74104) and reverse transcribed into cDNA using random hexamers and the SuperScript III First-Strand Synthesis System (Thermo Fisher Scientific, 18080051). The cDNA was used to prepare triplicate reactions for qRT-PCR according to manufacturer's instructions (QuantiTect SYBR Green PCR kit, Qiagen, 204143), plates were centrifuged shortly and run on a LightCycler 480 (Roche) using the following PCR conditions: denaturation for 15 s at 94℃; annealing for 30 s at 60℃; extension for 30 s at 72℃. The levels for each condition were corrected with their own input. The following primers were used for qPCR:

*Xenopus 18S rRNA*, 5'-GTAACCCGCTGAACCCCGTT-3' and 5'-CCATCCAATCGGTAGTAGCG-3';
*Xenopus 28S rRNA*, 5'-CTGTCAAACCGTAACGCAGG-3' and 5'-CTGACTTAGAGGCGTTCAGTCA-3'.
*human 18S rRNA*, 5'-GTAACCCGTTGAACCCCATT-3' and 5'-CCATCCAATCGGTAGTAGCG-3';
*human 28S rRNA*, 5'-AACGGCGGGAGTAACTATGA-3' and 5'-TAGGGACAGTGGGAATCTCG-3'.
*Xenopus ctnnb1 mRNA*, 5'-GACCACAAGTCGGGTGCTTA-3' and 5'- CCAGACGTTGGCTTGAGTCT-3';
*Xenopus hnrnph1 mRNA*, 5'- GGTTGGAAAATCGTGCCAAATG-3' and 5'- GCCTTTTCAGCTATTTCCTGTGAAG-3';
*Xenopus rps14 mRNA*, 5'- GTGACTGACCTGTCTGGCAA-3' and 5'- GCAACATCTTGTGCAGCCAA-3'.

## Proximity ligation assay

These experiments were carried out according to the manufacturer's protocol (Sigma-Aldrich, Duolink Biosciences) using Duolink In Situ Detection reagents (Sigma-Aldrich, DUO90214 or DUO92008). After 24 hr, cultures were fixed in 2% formaldehyde/7.5% sucrose in PBS for 20 min at RT, washed three times in PBS with 0.001% Triton-X-100, permeabilized for 5 min in 0.1% Triton-X-100 in PBS, washed three times in PBS with 0.001% Triton-X-100, blocked with 5% heat-inactivated goat serum in PBS for 45 min at RT and subsequently incubated with primary antibodies overnight at 4℃. Primary antibodies were diluted at 1:100 for mouse anti-DCC (BD Biosciences, 554223), 1:100 mouse-anti-EphB2 (Thermo Fisher Scientific, 37–1700) 1:100 for rabbit anti-RPL5/uL18 (Proteintech, 15430–1-AP), 1:100 rabbit anti-RPS4X/eS4 (Proteintech, 14799–1-AP), 1:100 rabbit-anti RPL10A/uL1 (Proteintech, 16681–1-AP), 1:100 for rabbit anti-neuropilin-1 (Abcam, ab81321), 1:100 mouse anti-RPS3A/eS1 (Abcam, ab194670),1:100 mouse-anti-RPS23/uS12 (Abcam, ab57644), rabbit-anti-hnRNPA2B1 (Abcam, ab31645), rabbit-IgG isotype control (Abcam, ab37415), mouse IgG1 isotype control (MAB002, R and D Systems). After primary antibody incubation, dishes were washed twice for 5 min with 0.002% Triton X-100 in PBS and incubated with anti-rabbit-PLUS (Sigma-Aldrich, DUO92002) and anti-mouse-MINUS (Sigma-Aldrich, DUO92004) PLA probes for 1 hr at 37℃, with ligase for 30 min at 37℃ and with the polymerase mix with red fluorescence for 100–140 min at 37℃. The samples were subsequently mounted with the mounting medium (DUO82040, Duolink) and imaged using a Nikon Eclipse TE2000-U inverted microscope equipped with an EMCCD camera. The number of discrete fluorescent puncta from randomly selected isolated growth cones were counted using Volocity software (Perkin Elmer).

## Immunocytochemistry

After 24 hr, *Xenopus* retinal cultures were fixed in 2% formaldehyde/7,5% sucrose in PBS for 20 min at RT. For the puromycilation assay, 48 hr old cultures were used, eyes were manually removed and axons were treated with 10 µg/ml puromycin (Sigma-Aldrich, P8833) for 10 min at RT before fixation. The fixed cultures were then washed three times in PBS with 0.001% Triton-X-100, permeabilized for 5 min at RT in 0.1% Triton-X-100 in PBS, washed again for three time in PBS with 0.001% Triton-X-100 and blocked with 5% heat-inactivated goat serum in PBS for 45 min at RT. Primary antibodies were incubated overnight at 4℃, followed by Alexa Fluor-conjugated secondary antibodies for 60 min at RT in the dark. Cultures were mounted in FluorSave (Calbiochem, 345789). Primary antibodies were used at the following dilutions: 1:100 for mouse anti-DCC (BD Biosciences, 554223), 1:100 for rabbit anti-neuropilin-1 (Abcam, ab81321), 1:100 for mouse-anti-neuropilin-1 (Proteintech, 60067–1-Ig), 1:100 for rabbit anti-RPL5/uL18 (Proteintech, 15430–1-AP), 1:100 mouse anti-RPS3A/eS1

(Abcam, ab194670), 1:200 mouse-anti-puromycin-AlexaFluor-488 (Millipore, MABE343-AF488), rabbit-anti-Staufen1 (Abcam, ab73478), rabbit-anti-hnRNPA2B1 (Abcam, ab31645), 1:500 rabbit-anti-β-Catenin (Sigma-Aldrich, C2206), 1:500 rabbit-anti-hnRNPH1 (Abcam, ab154894), rabbit-anti-RPS14/uS11 (Abcam, ab174661), 1:250 rabbit-anti-pERK1/2 (Cell Signaling, 9101). Secondary antibodies were diluted at: 1:1000 goat anti-rabbit Alexa Fluor 568 (Abcam, ab150077), 1:1000 goat anti-mouse Alexa Fluor 568 (Abcam, ab150117).

## Expansion microscopy

For expansion microscopy, RGCs explant cultures were immunostained with primary and secondary antibodies as described above, followed by applying the expansion protocol for cultured cells (*Chen et al., 2015*). Briefly, cultures were incubated in 0.25% glutaraldehyde in PBS for 20 min at RT and then washed with PBS three times, before adding monomer solution (2M NaCl, 8.625% (w/w) sodium acrylate, 2.5% (w/w) acrylamide, 0.1% (w/w) N,N'-methylenebisacrylamide in PBS) for 2 min at RT. Subsequently, monomer solution was mixed with 0.2% ammonium persulfate (APS) and 0.2% Tetramethylethylendiamin (TEMED) and added to the samples. Gelation of the polymer occurred at 37°C for 30 min, followed by digestion of the samples with digestion buffer (40 mM Tris (pH 8), 1 mM EDTA, 0.5% Triton-X-100, 0.8M guanidine NaCl, 8 U/ml Proteinase K in water) and incubated at 37°C for 1 hr. To expand the samples, digestion buffer was removed and gels were placed in water for several hours during which water was replaced every 30 min. Once gels detached from the glass dish, they were transferred to a bigger dish to allow expansion. For imaging, expanded gels were cut in pieces and transferred to poly-L-lysine coated glass bottom dishes. Imaging was performed using a 60x/1.3 NA silicone oil objective lens on a Perkin Elmer Spinning Disk UltraVIEW ERS, Olympus IX81 inverted microscope and the Volocity software. Images were processed by using Fiji (NIH) and co-localisation analysis was carried out by using a purpose-written Matlab (The MathWorks) code. For co-localisation analysis, images were multiplied with a mask of a focused area of interest and the average background fluorescence was subtracted, before Pearson's correlation coefficients were computed.

## Quantification of immunofluorescence

For the quantification of fluorescence intensity, isolated growth cones were randomly selected with phase optics. For each experiment, the images were captured on the same day using the same gain and exposure settings and pixel saturation was avoided. Using Volocity software (Perkin Elmer), a region of interest (ROI) was defined by tracing the outline of each single growth cone using the phase image and the mean pixel intensity per unit area was measured in the fluorescent channel. The background fluorescence was measured in a ROI close to the growth cone that was free of debree or other axons and this was substracted from the mean fluorescence value of the growth cone. For the co-localization analysis of RBPs with receptors (*Figure 2E–F*), masks of the region of interest of each imaged growth cone were automatically generated using a code written in the wolfram language in Mathematica (https://wolfram.com/mathematica). For this code, training data was generated first by using hand traced outlines of 30 growth cones in two channel fluorescence images using ImageJ (http://imagej.net) to generate 30 corresponding binary growth cone maps. We chose the U-Net architecture (*Ronneberger et al., 2015*) to learn the growth cone segmentation similar as done in *Jakobs et al. (2019)*. For training, we split the dataset into 25 training images and five validation images and down sampled every image so that the short dimension was 600 pixels long. During training *input* images were heavily augmented to prevent overfitting by (i) random cropping to 256 × 256 pixel sizes, (ii) random rotations, (iii) random reflections, (iv) random background gradients, (v) random noise, (vi) random nonlinear distortions. U-Net was with batch size eight and cross entropy loss until the validation loss did not decrease any further for 10 consecutive epochs on a nVidia 1080 Ti. The best performing network (using intersection over union benchmarking) was subsequently chosen to generate growth cone masks for our data. Masks were generated by first applying the best U-Net to the downsampled image followed by upsampling. The resulting output images were binarized by a morphological binarization algorithm with foreground threshold 0.3 that treats any pixel that is connected to the foreground and has a value larger than 0.2 also as part of the foreground.

## Mass-spectrometry

1D gel bands were transferred into a 96-well PCR plate. The bands were cut into 1 mm² pieces, destained, reduced (DTT) and alkylated (iodoacetamide) and subjected to enzymatic digestion with chymotrypsin overnight at 37˚C. After digestion, the supernatant was pipetted into a sample vial and loaded onto an autosampler for automated LC-MS/MS analysis.

All LC-MS/MS experiments were performed using a Dionex Ultimate 3000 RSLC nanoUPLC (Thermo Fisher Scientific Inc, Waltham, MA, USA) system and a Q Exactive Orbitrap mass spectrometer (Thermo Fisher Scientific Inc, Waltham, MA, USA). Separation of peptides was performed by reverse-phase chromatography at a flow rate of 300 nL/min and a Thermo Scientific reverse-phase nano Easy-spray column (Thermo Scientific PepMap C18, 2 µm particle size, 100A pore size, 75 µm i.d. x 50 cm length). Peptides were loaded onto a pre-column (Thermo Scientific PepMap 100 C18, 5 µm particle size, 100A pore size, 300 µm i.d. x 5 mm length) from the Ultimate 3000 autosampler with 0.1% formic acid for 3 min at a flow rate of 10 µL/min. After this period, the column valve was switched to allow elution of peptides from the pre-column onto the analytical column. Solvent A was water + 0.1% formic acid and solvent B was 80% acetonitrile, 20% water + 0.1% formic acid. The linear gradient employed was 2–40% B in 30 min.

The LC eluant was sprayed into the mass spectrometer by means of an Easy-Spray source (Thermo Fisher Scientific Inc). All m/z values of eluting ions were measured in an Orbitrap mass analyzer, set at a resolution of 70000 and was scanned between m/z 380–1500. Data-dependent scans (Top 20) were employed to automatically isolate and generate fragment ions by higher energy collisional dissociation (HCD, NCE:25%) in the HCD collision cell and measurement of the resulting fragment ions was performed in the Orbitrap analyser, set at a resolution of 17500. Singly charged ions and ions with unassigned charge states were excluded from being selected for MS/MS and a dynamic exclusion window of 20 s was employed.

Raw data were processed using Maxquant (version 1.6.1.0) (*Cox and Mann, 2008*) with default settings. MS/MS spectra were searched against the *X. laevis* protein sequences from Xenbase (xlaevisProtein.fasta). Enzyme specificity was set to trypsin/P, allowing a maximum of two missed cleavages. The minimal peptide length allowed was set to seven amino acids. Global false discovery rates for peptide and protein identification were set to 1%. The match-between runs option was enabled.

## Label-free quantification (LFQ) analysis of proteomics data

To identify significant interactors, t-test-based statistics were applied on label-free quantification (LFQ) intensity values were performed using Perseus software. Briefly, LFQ intensity values were logarithmized (log2) and missing values were imputed based on the normal distribution (width = 0.3, shift = 1.8). Significant interactors of DCC or Nrp1 pulldowns compared to IgG pulldowns were determined using a two-tailed t-test with correction for multiple testing using a permutation-based false discovery rate (FDR) method.

## RNA-sequencing

RNA was isolated from immunoprecipitated samples from SH-SY5Y cells as described above using RLT buffer (Qiagen) containing β-mercaptoethanol and the RNeasy Mini kit (Qiagen) followed by in-column DNase I treatment to remove genomic DNA contamination. RNA quality was analysed using Agilent RNA 6000 Pico kit and reagents (Agilent, 5067–1514,1535,1513) on a Agilent 2100 Bioanalyzer (Agilent). cDNA was then amplified using a method developed for single cell transcriptomics (*Tang et al., 2009*) with minor modifications (*Shigeoka et al., 2016*). The cDNA library preparation was performed using a KAPA Hyperprep kit (Roche) and cDNA libraries were subjected to a RNA-sequencing run on a Next-seq 500 instrument (Illumina) using the 150 cycles high output kit (Illumina).

## Bioinformatic analysis of RNA-sequencing data

The sequence reads were mapped using HISAT 2 version 2.1.0, and FPKM values were estimated using Cufflinks version 2.2.1. Read counts for each gene were determined using HTSeq version 0.11.0. Differential expression analysis was performed using edgeR in R version 3.5.0 (FDR < 0.05). The GO enrichment analysis was performed using topGO version 2.32.0. The mRNA targets of RBPs

were obtained from previously published studies as listed in the main text. To analyse the enrichment of Staufen1 and hnRNPA2B1 targets, all RBP targets that showed a significant difference between DCC and Nrp1 pulldowns were first selected and the log2 fold change values between DCC and Nrp1 were used for a Mann-Whitney U test (Wilcoxon rank sum test).

### Electron microscopy of axonal growth cones

Cultured neurons were fixed at 37°C for 45 min in 2.5% glutaraldehyde, sodium cacodylate buffer 0.1M pH7.4 containing 2 mM CaCl2 and 2 mM MgCl2. Samples were post-fixed for 15 min at RT in 1% osmium and embedded in epoxy resin. Ultrathin sections were imaged with a ZEISS EM 912 microscope. Ribosomes were identified based on size and shape. To quantify the inter-ribosome distance, the center-to-center distance was measured using ImageJ. For axonal growth cones, ribosomes were selected that were located within 50 nm of the plasma membrane and the distance to its closest neighbor was quantified.

### Statistical analysis

All experiments were performed in at least three independent biological replicates unless explicitly stated otherwise. The order of data collection was randomized, and no data were excluded from analysis. Statistical analysis was performed using GraphPad Prism, R or MATLAB. Statistical tests used are described in the figure legends.

### Data availability

RNA-sequencing data associated with this manuscript has been deposited on the GEO database (identifier GSE135338). All proteomics data associated with this manuscript has been uploaded to the PRIDE online repository (identifier: PXD015650).

## Acknowledgements

We thank Nicola Lawrence, Caia Duncan (Juan Mata lab, Unversity of Cambridge), and Katrin Mooslehner for technical assistance and Fabrice Richard (PiCSL-FBI core facility, IBDM, CNRS, Aix-Marseille University; member of the France-BioImaging national research infrastructure (ANR-10-INBS-04)) for assistance with EM experiments. This work was supported by the Netherlands Organization for Scientific Research (NWO Rubicon 019.161LW.033) (MK), UK Engineering and Physical Sciences Research Council, EPSRC Grants (EP/L015889/1 and EP/H018301/1) and Wellcome Trust Grants (3-3249/Z/16/Z and 089703/Z/09/Z) (CFK), Wellcome Trust Grants (085314/Z/08/Z and 203249/Z/16/Z) and European Research Council Advanced Grant (322817) (CEH).

## Additional information

### Funding

| Funder | Grant reference number | Author |
|---|---|---|
| Netherlands Organisation for Scientific Research | Rubicon 019.161LW.033 | Max Koppers |
| EPSRC | EP/L015889/1 | Clemens F Kaminski |
| EPSRC | EP/H018301/1 | Clemens F Kaminski |
| Wellcome Trust | 3-3249/Z/16/Z | Clemens F Kaminski |
| Wellcome Trust | 089703/Z/09/Z | Clemens F Kaminski |
| Wellcome Trust | 085314/Z/08/Z | Christine E Holt |
| Wellcome Trust | 203249/Z/16/Z | Christine E Holt |
| European Research Council | Advanced Grant 322817 | Christine E Holt |

The funders had no role in study design, data collection and interpretation, or the decision to submit the work for publication.

## Author contributions

Max Koppers, Conceptualization, Data curation, Formal analysis, Supervision, Funding acquisition, Validation, Investigation, Visualization, Methodology, Writing—original draft, Project administration, Writing—review and editing; Roberta Cagnetta, Formal analysis, Investigation, Visualization, Methodology, Writing—review and editing; Toshiaki Shigeoka, Conceptualization, Formal analysis, Investigation, Methodology, Writing—review and editing; Lucia CS Wunderlich, Julie Qiaojin Lin, Investigation, Methodology; Pedro Vallejo-Ramirez, Software, Formal analysis; Sixian Zhao, Anaïs Bellon, Formal analysis, Investigation; Maximilian AH Jakobs, Data curation, Wrote a script to automatically detect axonal growth cones from microscopy images and create masks from these enabling imaging quantification, Wrote the methods part for this; Asha Dwivedy, Performed in vitro retinal cultures and processed them for immunocytochemistry or PLA; Michael S Minett, Carried out in vitro retinal cultures, performed PLA and acquired imaging data; Clemens F Kaminski, Resources, Supervision; William A Harris, John G Flanagan, Conceptualization, Writing—review and editing; Christine E Holt, Conceptualization, Resources, Supervision, Funding acquisition, Writing—review and editing

## Author ORCIDs

Max Koppers ⓘD https://orcid.org/0000-0002-7751-1082
Lucia CS Wunderlich ⓘD http://orcid.org/0000-0001-7200-1713
Pedro Vallejo-Ramirez ⓘD https://orcid.org/0000-0002-7879-6761
Julie Qiaojin Lin ⓘD https://orcid.org/0000-0002-2669-6478
Maximilian AH Jakobs ⓘD https://orcid.org/0000-0002-0879-7937
Clemens F Kaminski ⓘD http://orcid.org/0000-0002-5194-0962
William A Harris ⓘD http://orcid.org/0000-0002-9995-8096
Christine E Holt ⓘD https://orcid.org/0000-0003-2829-121X

## Ethics

Animal experimentation: All animal experiments were approved by the University of Cambridge Ethical Review Committee in compliance with the University of Cambridge Animal Welfare Policy. This research has been regulated under the Animals (Scientific Procedures) Act 1986 Amendment Regulations 2012 following ethical review by the University of Cambridge Animal Welfare and Ethical Review Body (AWERB) and under project license PPL80/2198.

## Decision letter and Author response

Decision letter https://doi.org/10.7554/eLife.48718.sa1
Author response https://doi.org/10.7554/eLife.48718.sa2

# Additional files

## Supplementary files

• Transparent reporting form

## Data availability

RNA-sequencing data associated with this manuscript has been deposited on the GEO database (identifier GSE135338). All proteomics data associated with this manuscript has been uploaded to the PRIDE online repository (identifier: PXD015650).

The following datasets were generated:

| Author(s) | Year | Dataset title | Dataset URL | Database and Identifier |
|---|---|---|---|---|
| Koppers M, Cagnetta R, Shigeoka T, Wunderlich LCS, Vallejo-Ramirez P, Qiaojin Lin J, Zhao S, Jakobs M, Dwivedy A, Minett MS, | 2019 | LC-MSMS of DCC and Neuropilin-1 immunoprecipitated samples from Xenopus Laevis brains | https://www.ebi.ac.uk/pride/archive/projects/PXD015650 | PRIDE, PXD015650 |

| Author(s) | | | |
|---|---|---|---|
| Bellon A, Kaminski CF, Harris WA, Flanagan JG, Holt CE | | | |
| Koppers M, Cagnetta R, Shigeoka T, Wunderlich LCS, Vallejo-Ramirez P, Qiaojin Lin J, Zhao S, Jakobs M, Dwivedy A, Minett MS, Bellon A, Kaminski CF, Harris WA, Flanagan JG, Holt CE | 2019 | Receptor-specific interactome as a hub for rapid cue-induced selective translation in axons | https://www.ncbi.nlm.nih.gov/geo/query/acc.cgi?acc=GSE135338 | NCBI Gene Expression Omnibus, GSE135338 |

The following previously published datasets were used:

| Author(s) | Year | Dataset title | Dataset URL | Database and Identifier |
|---|---|---|---|---|
| Lebedeva S, Jens M, Theil K, Schwanhaeusser B, Selbach M, Landthaler M, Rajewsky N | 2011 | Unstressed HeLa cells and ELAVL1/HuR knock down conditions: polyA RNA-Seq, small RNA-Seq, and PAR-CLIP | https://www.ncbi.nlm.nih.gov/geo/query/acc.cgi?acc=GSE29943 | NCBI Gene Expression Omnibus, GSE29943 |
| Martinez F J, Pratt GA, Van Nostrand EL, Batra R, Huelga SC, Kapeli K, Freese P, Chun SJ, Ling K, Gelboin-Burkhart C, Fijany L, Wang HC, Nussbacher JK, Broski SM, Kim HJ, Lardelli R, Sundararaman B, Donohue JP, Javaherian A, Lykke-Andersen J, Finkbeiner S, Bennett CF, Ares Jr M, Burge CB, Taylor JP, Rigo F | 2016 | HNRNPA2B1 regulates alternative RNA processing in the nervous system and accumulates in granules in ALS IPSC-derived motor neurons | https://www.ncbi.nlm.nih.gov/geo/query/acc.cgi?acc=GSE86464 | NCBI Gene Expression Omnibus, GSE86464 |
| Ascano M, Mukherjee N, Bandaru P, Miller JB, Nusbam J, Corcoran D, Langlois C, Munschauer M, Hafner M, Williams Z, Ohler U | 2012 | FMR1 targets distinct mRNA sequence elements to regulate protein expression | https://www.ncbi.nlm.nih.gov/geo/query/acc.cgi?acc=GSE39686 | NCBI Gene Expression Omnibus, GSE39686 |
| oichiro Sugimoto, Alessandra Vigilante, Elodie Darbo, Alexandra Zirra, Cristina Militti, Andrea D'Ambrogio, Nicholas M Luscombe | 2015 | hiCLIP analysis of RNA duplexes bound by STAU1 in HEK293 cells | https://www.ebi.ac.uk/arrayexpress/experiments/E-MTAB-2937/ | ArrayExpress, E-MTAB-2937 |

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
