## [Decision Letter]

**Acceptance summary:**

Your paper is a welcome follow-up to John Flanagan's previous study that showed that ribosome and mRNA-protein complexes assemble at the cytoplasmic tail of DCC, to provide translational specificity upon ligand binding. Your demonstration that unique RBPs and mRNAs associate with Neuropilin1, Robo2 and DCC but not EphB2 expands upon this process. The receptor-specific interactomes you describe provide an excellent model for the control of cue-specific local translation and will be appreciated by those who work on molecular mechanisms of the receptors and ligands formerly thought to function strictly in axonal growth and guidance but are now implicated in synaptogenesis, vasculogenesis and tumor regulation.

**Decision letter after peer review:**

[Editors’ note: this article was originally rejected after discussions between the reviewers, but the authors were invited to resubmit after an appeal against the decision.]

Thank you for submitting your work entitled "Receptor-specific interactome as a hub for rapid cue-induced selective translation in axons" for consideration by *eLife*. Your article has been reviewed by three peer reviewers, and the evaluation has been overseen by a Reviewing Editor and a Senior Editor. The reviewers have opted to remain anonymous.

Our decision has been reached after consultation amongst the reviewers. Based on these discussions and the individual reviews below, we regret to inform you that your work is not acceptable for publication in *eLife* at this time.

All three reviewers had high praise for this study done in collaboration with John Flanagan's group, that ribosomes can couple with surface receptors in subcellular compartments, and that specific RBPs recruit distinct subsets of mRNAs to these receptors, to provide a potential mechanism of how specificity of translation is achieved following an extracellular cue. That the Semaphorin receptor Neuropilin-1 in addition to DCC, but not EphB2, is associated with ribosomal proteins and RNAs, as well as with RNA-binding proteins and mRNAs, is welcome news. The work is exciting and we applaud your execution of exceedingly difficult analyses.

However, as you can read in the appended reviews, the reviewers were critical of a number of aspects and call for amendments, several of them non-overlapping, that should be addressed before publication. The key categories of requested amendments include:

- You report a vast number of mRNAs associated with DCC and Nrp1. In the consultation session among reviewers, and from reviewer 2's critique, they wonder if there is a distinction between the dominant mRNAs and others at low levels, questioning the relevance of the latter. Reporting the raw data/data files for the proteomics and RNA-seq experiments would be essential in your revision, for the reader to have access to the full list.

- The authors assume that the binding of guidance cues to their respective receptors leads to dissociation of the ribosomes and subsequent local translation. They ask for evidence that this is the case and that the transcripts are translated into proteins.

In line with this critique, reviewer 3 states that assuming that the RNAseA/T1 treatment effectively shows that mRNA/protein interactions are needed for the ribosome subunits to interact with DCC and Nrp1, they wondered whether you could determine whether RNA is needed for the RBP associations with these receptors, and similarly whether ribosome subunits association is needed for mRNA interaction with DCC and Nrp1.

Reviewer 2 queries whether the reported association takes place on vesicles (either during the transport from the Golgi or upon endocytosis) as per the Holt lab's study 'Late Endosomes Act as mRNA Translation Platforms' (Cioni et al., 2019).

- While reviewer 1 commends you on the striking and convincing EM images of ribosomes within growth cones, one reviewer in the consultation session wondered whether your point on ER and Golgi association could be sorted out by FISH and that including such an analysis for at least a few key mRNAs would be helpful.

- Finally, the reviewers all had comments on rigor/quantification for replication's sake: several figures lack unbiased analyses and biological replication, and there are numerous single, non-repeated experiments.

Reviewer #1:

In this interesting study the authors provide evidences that ribosomes can couple with surface receptors in subcellular compartments, allowing a tight spatiotemporal control of local translation in response to extracellular stimuli. The authors report that this is indeed not a limited phenomenon and show that several receptors whose activation leads to local translation are coupled with ribosomes. Moreover, specific RBPs recruit distinct subsets of mRNAs to these receptors, thereby providing a potential mechanism explaining how specificity of translation is achieved following an extracellular cue. The study is novel and interesting, and most experiments are well controlled. However, there are some issues that need to be addressed before this would be suitable for publication.

- One of the main findings of this study is that specific populations of mRNAs are targeted to different receptors through interactions with different RBPs. However, it is unclear whether these interactions are maintained in the absence of stimulation or during stimulation paradigms that do not lead to ribosomal dissociation, such as EphrinA1+Netrin1 stimulation of the DCC receptor.

- The authors suggest that the binding of guidance cues to their respective receptors leads to dissociation of the ribosomes and subsequent local translation. They should provide evidence that this is the case and that the transcripts are translate into proteins.

- The controls for the co-sedimentation polysome experiments in Figure 2—figure supplement 1 are weak. EDTA slightly reduces the levels of receptors identified in the heavier fractions, but this may be due to different exposure, as there seems to be an overall decrease in total levels of protein.

- For Figures 2B and 2C the authors should show the input levels for Rps3A and Rps26 to demonstrate that the treatment leading to ribosomal dissociation doesn't induce an overall reduction of these components, especially given that the binding with the receptor is quite limited.

Reviewer #2:

In this manuscript, the authors follow up on a 2010 paper by the Flanagan group (Tcherkezian et al.) that reported the association of ribosomal subunits with the intracellular end of the Netrin-1 receptor DCC. Here, Koppers et al. describe that not only DCC but also the Semaphorin receptor Neuropilin-1, but not EphB2 are associated with ribosomal proteins and RNAs, as well as with RNA-binding proteins and mRNAs. The ribosomes and RNAs dissociate from the receptors upon stimulation with Netrin-1 or Sem3A, respectively, and in the case of DCC, this effect is blocked by co-stimulation with EphrinA1. Together, the authors propose that receptor-specific interactomes provide a model for the control of cue-specific local translation.

The findings reported in this manuscript would be of interest to neurobiologists and would represent a significant advance in the understanding of the control of local protein synthesis in developing axons. However, several key aspects of the manuscript are currently underdeveloped, there are unaddressed conceptual questions, and several figures are problematic due to lacking unbiased analyses and biological replications.

Conceptual questions:

The authors find an astonishingly great number of mRNAs associated with DCC and Nrp1. Just the differentially associated mRNAs are 158 and 383, respectively. The total number is not even reported. How would these numbers be possibly compatible with the proposed model of 'selective' control of local translation? Does every receptor associate with several hundred mRNAs and all of them get translated upon ligand binding? If conversely every receptor is associated with just a couple of mRNA species, how can this model ensure reliable translational responses to ligand binding?

The authors focus entirely on DCC and Nrp1 on the cell surface. Would it not be possible that the reported association takes place on vesicles (either during the transport from the Golgi or upon endocytosis)? After all, the same group just published that 'Late Endosomes Act as mRNA Translation Platforms' (Cioni et al., 2019). The finding of vesicle-mediated transport as a category for interactors of DCC and Nrp1 seems to support this alternative idea.

Translational targets for Netrin-1 and Sema3A have been described by the authors and others but are being ignored in this manuscript. Are the mRNAs coding for β-actin, Par-3, Tctp associated with DCC, or is RhoA mRNA associated with Nrp-1? If they are not part of the interactomes, what does this mean for the proposed model?

Other major points in order of occurrence:

Figure 1G-J: The presented western blots are single experiments without replication and are not quantified. The presentation of single, non-repeated experiments is a recurrent problem in this manuscript.

Figure 1—figure supplement 1F, G: RPL39 is mentioned in the main text, but does not appear in the figure. The variances in this figure are quite large, and as the authors state, the conclusion is preliminary. The authors should either perform more experiments to elevate the conclusion from preliminary or remove this entire figure/chapter.

Figures 2—figure supplement 1A, B: Single, unrepeated experiments without a proper quantification. This is especially troublesome, as the purported shift for DCC upon EDTA treatment is not obvious at all.

Figure 2A: The RNAseA/T1 treatment is an elegant approach but to follow the authors interpretation it would be critical to experimentally prove that the rRNAs stay intact.

Figures 2D, E: The author present the same plots as in Figure 1A without mentioning this fact. Also, there is no congruency between the main text, the figure legend and the actual figure: where are the 10 shared RBPs?

Figure 2—figure supplement 1C, D: Single unrepeated, non-quantified experiments.

Figures 3A, B: Single growth cones are presented without replication, quantification or any unbiased co-localization analysis.

Figure 3E: Why did the authors normalize the data instead of presenting the actual counts per sq. mm?

Figure 3F: The authors are correct by stating that association of DCC with RPL5 and RPS4X is significantly decreased, however, they ignore the vastly different effect sizes. RPL5 seems reduced by ~80% while RPS4X is reduced by less than 20%. What is the explanation for this and shouldn't the reduction of the ribosomal subunits be stoichiometrically matched to form functioning ribosomes? The authors argue that local production of RPLs might obscure their results; they should perform the experiments in the presence of CHX to excluded this potentially confounding possibility.

Figure 4A: It is suprising to see that Netrin-1 does not induce local translation as measured by puromycylation. How do the author reconcile this finding with previous reports that Netrin-1 induces local translation?

Figure 5A: A single image without biological repeats or (unbiased) quantification.

Reviewer #3:

This manuscript follows up on previous work from the Flanagan lab showing that ribosome and mRNA-protein complexes assemble at the cytoplasmic tail of DCC, to provide translational specificity upon ligand binding. Koppers et al. show that unique RBPs and mRNAs associate with Neuropilin 1, Robo2 and DCC but not EphB2. This brings an appealing mechanism to modulate specificity of translational regulation in response to extracellular stimuli, and the data are strengthened by lack of RNA and ribosome association EphB2 fits since its output is reportedly protein synthesis independent. Overall, the work is very well done, the data are provocative, and I think this represents an advance appropriate for *eLife*. However, there are some weaknesses that detract from my enthusiasm.

1) The differential association of specific RPs with DCC vs. Neuropilin-1 is intriguing and distinct ribosome populations have been suggested in a number of systems but unequivocally proven. Some of the techniques used here and methodologies rely on those in the Shigeoka et al., 2018 reference from the authors. Particularly, the experiments with RNAseA/T1 in Figure 2. The reference there is incomplete and I do not see it has been published except for bioRxiv (I am not certain on *eLife*'s policy for referencing unreviewed work like this).

2) Assuming the RNAseA/T1 treatment effectively shows that mRNA/protein interactions are needed for the ribosome subunits to interact with DCC and Nrp1, it is surprising that no efforts are made to determine whether RNA is needed for the RBP associations with these receptors. Similarly whether ribosome subunits association is needed for mRNA interaction with DCC and Nrp1.

3) The MS data for different RPs in Figure 1—figure supplement 1 is surprising and does support the authors' suggestion for different ribosome populations. It is surprising that the authors did not validate those showing greatest differentials (e.g., L4, L35, S6 and S9). Also, it seems like there are more 60S than 40S RPs showing differential association, but the authors did not comment on this point.

4) For Figure 2—figure supplement 1A-B, the majority of Nrp1 and DCC signals do not fractionate with subunits. Granted, these are very hard experiments and the EDTA treatments do shifts signals for Nrp1 and DCC upwards in the gradients. I think this would be strengthened EDTA treatments and some quantitation for the colocalization data shown in Figure 3 and Figure 3—figure supplement 1.

5) It is surprising that no colocalization for mRNAs with the receptors is provided beyond the co-IP, and such RIP experiments are known to have artefacts from in solution RNA protein-interactions after lysis.

6) For Figure 5A, the EM is compelling and shows a surprisingly high number of electron dense ribosome like structures. The authors do not note whether this is a stimulated or naïve culture. Comparison of the two would be informative and strengthen the authors' conclusions (though I do recognize the difficulty in this request).

7) Robo3.3 and EphA2 have been shown to be translated in axons. This is probably worth mentioning in the Discussion since the authors show that EphB2 does not use this mechanism for sequestering RNPs and ribosome-bound mRNAs.

[Editors’ note: what now follows is the decision letter after the authors submitted for further consideration.]

Thank you for resubmitting your work entitled "Receptor-specific interactome as a hub for rapid cue-induced selective translation in axons" for further consideration by *eLife*. Your revised article has been evaluated by Catherine Dulac as the Senior Editor, and a Reviewing Editor.

The manuscript has been improved and should be a welcome addition to the work of your laboratory and John Flanagan's group on local translation, in particular, that ribosomes can couple with specific RBPs that in turn recruit distinct subsets of mRNAs, to provide a potential mechanism of how specificity of translation is achieved following an extracellular cue.

Two of the reviewers found your manuscript satisfactorily amended, but one of the reviewers cited a number of points that should be addressed before publication, all of them addressed by textual amendments:

1) There are a number of places where your results are somewhat overstated and should be toned down:

a) Abstract: 'Our findings.… provide a general model for the rapid, localized and selective control of cue-induced translation.' Compare this to the much more accurate and measured summary: '… this study provides evidence…and suggests…'. Your results are interesting and represent a significant advance, but it may be premature to consider that they comprise a general model.

b) The findings from Figure 2B, C are summarized: '…these results suggest that the interaction.… is likely mediated through mRNA'. One page later, this sentence has morphed into the definitive statement: '…our finding that mRNA mediates the association of receptors with specific ribosomes…'

c) '…reveal that multiple receptors.… can associate with ribosomes.' This statement is imprecise; the results up to this point show association with ribosomal subunits, not ribosomes.

Please carefully edit the Results section and relegate the interpretation especially statements regarding the general impact/significance, to the Discussion.

2) A major concern in the first review was that some experiments were not repeated and quantified. Most of these issues have been fixed, but in Figure 1C-E, Figure 1—figure supplement 1A-C, results are still presented without quantification. The sentence in the figure legend that these experiments have been repeated three times but the actual results should be presented.

3) Figure 2—figure supplement 1C: the n for these experiments are 2 or 1. Showing bar graphs with SDs is not very meaningful for such low n numbers. The individual data points should be shown instead.

4) Subsection “DCC and Nrp1 bind to specific subsets of mRNAs”, first paragraph: The finding that high abundance mRNAs are much more differentially associated than low abundance mRNAs seems to indicate that these mRNAs are the mRNAs that conform to the authors proposed model. It is a missed opportunity to not talk more about these mRNAs: provide numbers, identity, GO analysis.

5) In response to conceptual question 2 (reviewer 2, first review), you provide the interesting finding that endocytosis is required for the dissociation between DCC and RP. However, the question was in relation to the mRNA-seq findings: it is my understanding that the RNA-seq experiments cannot distinguish between mRNAs associated with the receptors at the membrane vs. receptors at vesicles. For example, if mRNAs/RNPs are associated receptors on vesicles that transport the receptors into axons, these mRNAs would show up in the RNA-seq, even if they no longer associate with receptors after insertion into the plasma membrane. Please discuss this point to make clear that the RNA-seq results might include mRNAs that are not associated with receptors in the membrane.

6) The response to the comment regarding Figure 3F is unsatisfactory: You seem to indicate that the data as presented are unreliable due to problems with the reagents. Expand on this point.

---

## [Author Response]

[Editors’ note: the author responses to the first round of peer review follow.]

[…] As you can read in the appended reviews, the reviewers were critical of a number of aspects and call for amendments, several of them non-overlapping, that should be addressed before publication. The key categories of requested amendments include:- You report a vast number of mRNAs associated with DCC and Nrp1. In the Consultation session among reviewers, and from reviewer 2's critique, they wonder if there is a distinction between the dominant mRNAs and others at low levels, questioning the relevance of the latter. Reporting the raw data / data files for the proteomics and RNA-seq experiments would be essential in your revision, for the reader to have access to the full list.

We have deposited the (raw) data files from our RNA-sequencing experiments on the GEO database (GSE135338). The proteomics raw data files have been deposited on PRIDE (PXD015650). This information has been added to the Data availability section.

The reviewers raise a good point about dominant mRNAs. Therefore, we have performed further bioinformatic analysis to look at any differences between dominant versus low abundant mRNAs. This showed that the majority (67.9%) of high abundant mRNAs (FPKM > 1000) are differential mRNAs between DCC and Nrp1 while less than 5% of low abundant mRNAs (FPKM 1-10) are differential. This is interesting as it indicates that different receptors tend to associate differentially with more dominant mRNAs and suggests that these mRNAs are most relevant for cue-induced selective translation. We have included this in the subsection “DCC and Nrp1 bind to specific subsets of mRNAs”.

- The authors assume that the binding of guidance cues to their respective receptors leads to dissociation of the ribosomes and subsequent local translation. They ask for evidence that this is the case and that the transcripts are translated into proteins.

We find this comment puzzling because we did provide direct evidence for both the cue-induced receptor-ribosome dissociation and subsequent translation in axons in the original submitted manuscript.

Original Figure 3 shows that cue stimulation leads to receptor-ribosome dissociation for both DCC and Nrp1 in axonal growth cones, using proximity-based assays (Figure 3H-I). An important further question, one which the reviewers did not raise but is related to this issue, is to ask whether the receptor-ribosome dissociation is cue-specific. We have now conducted a new set of experiments to investigate this and we find that the dissociation is, indeed, highly cue-specific: Netrin-1 causes dissociation of ribosomal proteins (RPs) from DCC, but not Nrp1, whereas Sema3A causes dissociation of RPs from Nrp1, but not DCC. We have added these new data in Figure 3J-K and related text in the third paragraph of the subsection “Dissociation of ribosomes from receptors is triggered by extrinsic cues and requires endocytosis”.

Regarding the second point, whether receptor-associated mRNAs are locally translated into proteins on stimulation, we provided direct evidence in the original manuscript that this, indeed, is the case.

Figure 4E-H shows the validation of the protein changes in response to Netrin-1 but not Sema3A, for two corresponding mRNAs (*ctnnb1* and *hnrnph1*) that are identified as significantly enriched after DCC versus Nrp1 pulldown in our RNA-seq dataset, consistent with our previously reported cue-induced nascent proteome data (Cagnetta et al., 2018). In addition, Figure 4—figure supplement 1C-E shows the association of another mRNA – *rps14* – with DCC in *Xenopus* brains and the cue-specific protein increase of its corresponding protein in response to Netrin-1 but not Sema3A, again consistent with our previous proteomic data (Cagnetta et al., 2018). We have now made the text clearer (subsection “Integration of multiple cues can affect the cue-induced selective translation of 332 receptor-specific mRNAs”, third paragraph) stating that β-catenin, hnRNP-H1 and RPS14 have been detected in our recent axonal nascent proteome study to be selectively translated in response to Netrin-1 but not Sema3A (Cagnetta et al., 2018), and we also specified that their corresponding mRNAs have been detected in retinal axons by RNA-seq (Shigeoka et al., 2018). We hope that this clarifies any misunderstanding.

In line with this critique, reviewer 3 states that assuming that the RNAseA/T1 treatment effectively shows that mRNA/protein interactions are needed for the ribosome subunits to interact with DCC and Nrp1, he/she wondered whether you could determine whether RNA is needed for the RBP associations with these receptors, and similarly whether ribosome subunits association is needed for mRNA interaction with DCC and Nrp1.

To address these questions, we have now quantified the effect of RNAseA/T1 treatment on the binding between Nrp1 and the RNA-binding protein (RBP), Staufen1. The analysis shows a small, but significant decrease in Staufen1 binding to Nrp1 (Figure 2—figure supplement 1E). The decrease is smaller than that seen for ribosomal proteins but, nonetheless, suggests that mRNA can contribute to stabilizing the interaction between receptors and RBPs.

Additionally, to show our treatments work effectively, we have now added data from sucrose density gradients showing that our RNAseA/T1 treatment effectively cleaves unprotected mRNA but leaves rRNA intact (Figure 2—figure supplement 1D and subsection “Guidance cue receptors associate with ribosomes in a mRNA-dependent manner”, last paragraph).

The question about whether ribosome subunit association is needed for the association of mRNA with receptors is difficult to answer experimentally because of the lack of a clean way to dissociate ribosome subunits. We have tried to address this question by performing qPCR quantification of β-actin mRNA after DCC pulldown with and without EDTA treatment (which decreases the association of ribosome subunits with DCC). EDTA treatment resulted in a loss of β-actin mRNA binding to DCC suggesting that ribosome binding is needed for the mRNA interaction with DCC. Because the EDTA treatment may have affected the mRNA-DCC interaction directly or the DCC-RBP interaction it seems hard to draw a definitive conclusion from this experiment so we have not included it in the current manuscript, although we could if the reviewer and editor feel this is important.

Reviewer 2 queries whether the reported association takes place on vesicles (either during the transport from the Golgi or upon endocytosis) as per the Holt lab's study 'Late Endosomes Act as mRNA Translation Platforms' (Cioni et al., 2019).

This is an excellent question. We argue that the enrichment of the functional group “vesicle-mediated transport” associated to the receptors (Figure 1B), highlighted by the reviewer, may likely serve to mediate the endocytosis of the receptor upon cue stimulation (e.g. Nrp1 endocytosis upon Sema3A stimulation; (Zylbersztejn et al., 2012). Netrin-1 and Sema3A are indeed well known to cause rapid (1-2 min) endocytosis of their respective receptors, DCC and Nrp1, and inhibition of such endocytosis blocks local protein synthesis in response to ligand (Konopacki et al., 2016; Piper et al., 2005). Therefore, it is possible that the dissociation of the ribosome from the receptor also depends on endocytosis. For this reason, we have now performed new experiments to assess whether endocytosis is required for cue-induced receptor-ribosome dissociation by using Dynasore, an endocytosis inhibitor. The results show that inhibition of endocytosis completely blocks the cue-induced DCC-RP dissociation (now Figure 3L).

Furthermore, our focus in this manuscript concerns the role of receptor-ribosome interactions in responses to extracellularly applied ligand in vitro, leading to a protein synthesis response. Guidance molecules are also present extracellularly in vivo along the retinotectal pathway, therefore, we reason that the interaction between the ligand and the receptor is likely to occur at the cell surface.

In addition to this, we have performed new analyses on EM images obtained from axonal growth cones which show that 20 out of 22 axonal growth cones contain rows of ribosomes aligned just under the plasma membrane (within 50nm), again supportive of our model (subsection “Receptor-ribosome coupling occurs in RGC axonal growth cones”). We have added these new data and analyses in Figure 3F, 3G and Figure 3—figure supplement 1C-E.

- While reviewer 1 commends you on the striking and convincing EM images of ribosomes within growth cones, one reviewer in the Consultation session wondered whether your point on ER and Golgi association could be sorted out by FISH and that including such an analysis for at least a few key mRNAs would be helpful.

We are pleased that reviewer 1 liked our EM image of ribosomes in growth cones. To address the concern raised here, we have now added additional EM images as well as new quantitative analysis on these EM images (Figure 3F-G and Figure 3—figure supplement 1C-E) which are discussed in more detail below.

We are not sure exactly what is being referred to here regarding “…your point on ER and Golgi association…” since we do not mention ER or Golgi in our manuscript. The possible involvement of ER and Golgi in local translation in axons is currently controversial as ultrastructural evidence of canonical rough (ribosome-containing) ER or Golgi in axons is unclear (Gonzalez et al., 2018). Furthermore, it is important to emphasize that what we are focusing on here is evidence that different extracellular cues regulate the interaction of their specific receptors with ribosomes, and the subsequent translation of different sets of associated mRNAs. Therefore, while it is an interesting possibility that ER and Golgi components in axons are, indeed, involved in local translation, we believe that investigating this properly would constitute a further full study on its own and argue that this is beyond the scope of the current study. In the meantime, we prefer to refrain from speculations on this topic.

- Finally, the reviewers all had comments on rigor/quantification for replication's sake: several figures lack unbiased analyses and biological replication, and there are numerous single, non-repeated experiments.

We sincerely apologize for the omission in the presentation of the original manuscript. All of the experiments were repeated several times with biological replicates. As discussed in more detail below, where suitable we have also added unbiased analyses. We have now clarified this in the figure legends and Materials and methods section.

Reviewer #1:[…]- One of the main findings of this study is that specific populations of mRNAs are targeted to different receptors through interactions with different RBPs. However, it is unclear whether these interactions are maintained in the absence of stimulation or during stimulation paradigms that do not lead to ribosomal dissociation, such as EphrinA1+Netrin1 stimulation of the DCC receptor.

The reviewer raises an interesting point, the initial receptor-IP-MS was done on whole brains/eyes where the activation state of the receptors is unknown. For the RNA-seq experiments, we used a human neuronal cell line, showing that Nrp1 and DCC associate with different mRNAs in the absence of stimulation (Figure 2G, Figure 2—figure supplement 1F-I). Unfortunately, it is not possible to perform receptor-IPs, followed by RNA-seq analysis, on axons in response to different ligands because of the limited amount of axon-only material.

While the use of different systems was mentioned in the original manuscript, we have now clarified these points further in the subsections “DCC and Nrp1 bind to specific subsets of mRNAs” and “Integration of multiple cues can affect the cue-induced selective translation of receptor-specific mRNAs”.

- The authors suggest that the binding of guidance cues to their respective receptors leads to dissociation of the ribosomes and subsequent local translation. They should provide evidence that this is the case and that the transcripts are translate into proteins.

We were puzzled by this comment as we provided direct evidence that cue stimulation leads to receptor-ribosome dissociation in axonal growth cones (Figure 3H-I). The same conclusion was previously reported for DCC (Tcherkezian et al., 2010). We have now also added new control experiments showing that stimulation with the receptor-specific guidance cue is needed for receptor-ribosome dissociation, highlighting the specificity of these events. We have added these new data in Figure 3J-K and in the third paragraph of the subsection “Dissociation of ribosomes from receptors is triggered by extrinsic cues and requires endocytosis”.

In Figure 4E-H and Figure 4—figure supplement 1D-E we show that hnRNPH1, β-catenin (whose mRNAs were identified as significantly enriched after DCC versus Nrp1 pulldown in our receptor IP-RNA-seq analysis) and RPS14 increase upon Netrin-1, but not Sema3A, stimulation. Importantly, this is consistent with our previous proteomic-based study which showed that hnRNPH1, β-catenin and RPS14 are axonally translated in response to Netrin-1 but not Sema3A stimulation (Cagnetta et al., 2018), and we previously found that their transcripts are present in RGC axons (Shigeoka et al., 2018). Therefore, we do provide evidence that guidance cue stimulation leads to receptor-ribosome dissociation and that receptor-bound mRNA transcripts are translated into proteins after cue stimulation in axons. We have now made this clearer in the subsection “Integration of multiple cues can affect the cue-induced selective translation of receptor-specific mRNAs”.

- The controls for the co-sedimentation polysome experiments in Figure 2—figure supplement 1 are weak. EDTA slightly reduces the levels of receptors identified in the heavier fractions, but this may be due to different exposure, as there seems to be an overall decrease in total levels of protein.

We performed these experiments to provide further evidence in support of the association of receptors with ribosomes. This is in addition to evidence that we provide by: i) IP followed by Western blot and mass spectrometry (Figure 1A-F), ii) IP followed by qPCR (Figure 1G-J), iii) expansion microscopy imaging (Figure 3A-B), iv) proximity ligation assay (PLA) (Figure 3C-D). It is noteworthy that these polysome EDTA experiments were also performed for DCC in Tcherkezian et al., 2010, and our results are consistent with their findings. Furthermore, the results shown in Figure 2A-C strongly support that EDTA treatment results in loss of association of receptors to ribosomes, consistent with these polysome experiments.

Care was taken to use the same amount of starting material for these experiments to rule out differences in total protein levels. Control and EDTA treated samples were compared directly, twice for DCC and once for Nrp1. We then quantified the relative amounts of DCC (n = 2 for control and EDTA) and Nrp1 (n = 2 for control, n = 1 for EDTA) in ribosome-free and ribosome-containing fractions. This relative quantification corrects for any possible decrease in overall protein levels. These quantifications show a clear trend towards a shift to lighter fractions (i.e. ribosome-free fractions) for both DCC and Nrp1 after EDTA treatment (Figure 2—figure supplement 1C and subsection “Guidance cue receptors associate with ribosomes in a mRNA-dependent manner”). We hope that the reviewer appreciates that these data further support the association of DCC and Nrp1 with ribosomes. Lastly, to improve the presentation of the figure we have now aligned the UV profiles with the corresponding Western blot (Figure 2—figure supplement 1A-B).

- For Figures 2B and 2C the authors should show the input levels for Rps3A and Rps26 to demonstrate that the treatment leading to ribosomal dissociation doesn't induce an overall reduction of these components, especially given that the binding with the receptor is quite limited.

In these experiments, all the conditions come from the same starting material and thus have the same input levels. The samples were treated with EDTA and RNAseA/T1 only after pulldown so the results cannot be explained by an overall reduction in components. We followed this procedure based on Simsek et al., 2017, where they examine the RNA dependency of ribosomal protein interactors using this method. The fact that the pulled-down proteins (i.e. DCC and Nrp1) do not change is a good control with this method and quantification was corrected for the amount of pulled-down protein. We apologize for not explaining this clearly in the original manuscript and we have now clarified this in subsection “Guidance cue receptors associate with ribosomes in a mRNA-dependent manner” and the Materials and methods section.

Reviewer #2:[…]Conceptual questions:The authors find an astonishingly great number of mRNAs associated with DCC and Nrp1. Just the differentially associated mRNAs are 158 and 383, respectively. The total number is not even reported. How would these numbers be possibly compatible with the proposed model of 'selective' control of local translation? Does every receptor associate with several hundred mRNAs and all of them get translated upon ligand binding? If conversely every receptor is associated with just a couple of mRNA species, how can this model ensure reliable translational responses to ligand binding?

We have now deposited the (raw) data files from our RNA-sequencing experiments on the GEO database (GSE135338).

We detected several thousand mRNAs in the human cell line SH-SY5Y, with 100-400 that are differential between DCC and Nrp1. Cagnetta et al., 2018, showed that the translation of more than 100 mRNAs is regulated within 5 min in *Xenopus* retinal axons in response to Netrin and Sema3A, therefore, the numbers that we find in this manuscript do not seem very surprising, although differences in specific mRNAs or number of mRNAs between the two different systems may exist. This is exemplified by the absence of rps14 mRNA enrichment in SH-SY5Y cells, which was detected in *Xenopus* brain (Figure 4—figure supplement 1C).

More broadly, the conceptual question of how specificity of translation is achieved when there are so many mRNAs associated with each receptor is an interesting one. This question applies to many studies in the field, which have detected a large number of mRNAs in the axon. The answer may, in part, be that different mRNAs exist in common complexes (Buxbaum et al., 2015), but it clearly does not appear to be a simple one-receptor one-message pathway. It seems more probable that every receptor molecule exposed to the ligand has a certain probability of triggering the translation of a few associated mRNAs, and this could explain a consistent response of hundreds of mRNAs being translated when a particular ligand is added. In addition, the receptor-associated mRNAs were obtained from a cell line, as stated clearly in the original manuscript, so the number of receptor-associated mRNAs in axons may be different. Finally, it is interesting to speculate that a further fraction of the mRNAs detected may play a structural role, as recently reported in (Crerar et al., 2019). We have now addressed this conceptual point in the fourth paragraph of the Discussion.

The authors focus entirely on DCC and Nrp1 on the cell surface. Would it not be possible that the reported association takes place on vesicles (either during the transport from the Golgi or upon endocytosis)? After all, the same group just published that 'Late Endosomes Act as mRNA Translation Platforms' (Cioni et al., 2019). The finding of vesicle-mediated transport as a category for interactors of DCC and Nrp1 seems to support this alternative idea.

We thank the reviewer for raising this point. We argue that the enrichment of the functional group “vesicle-mediated transport” associated to the receptors (Figure 1B), highlighted by the reviewer, may likely serve to mediating the endocytosis of the receptor upon cue stimulation (e.g. Nrp1 endocytosis upon Sema3A stimulation; Zylbersztejn et al., 2012). Netrin-1 and Sema3A are indeed well known to cause rapid (1-2 min) endocytosis of their respective receptors, DCC and Nrp1, and inhibition of such endocytosis blocks local protein synthesis in response to ligand (Konopacki et al., 2016; Piper et al., 2005). Therefore, it is possible that the dissociation of the ribosome from the receptor also depends on endocytosis. For this reason, we have now performed new experiments to assess whether endocytosis is required for cue-induced receptor-ribosome dissociation using Dynasore, an endocytosis inhibitor. The results show that inhibition of endocytosis blocks the cue-induced DCC-RP dissociation (now Figure 3L), supporting our model that the receptor-ribosome complex is present at the cell surface.

Furthermore, we show that the dissociation of ribosomes from receptors happens in response to extracellularly applied ligand in vitro, leading to a protein synthesis response. This strongly supports the view that the interaction between the ligand and the receptor occurs at the cell surface in these in vitro experiments. Guidance molecules are also presented extracellularly in vivo along the retinotectal pathway. Therefore, it seems most likely that the initial interaction between ligand and receptor also occurs at the cell surface in vivo, and it is this ligand receptor interaction at the cell surface that triggers the dissociation of the receptor and the ribosome intracellularly. The subsequent intimate association between specific mRNAs and the ribosome may indeed require endocytosis and an endosomal platform.

In addition to this, we have performed new analyses on EM images obtained from axonal growth cones which show that 20 out of 22 axonal growth cones contain rows of ribosomes aligned just under the plasma membrane (within 50nm), again supportive of our model. We have added these new results in Figure 3F, 3G and Figure 3—figure supplement 1C-D and in the subsection “Receptor-ribosome coupling occurs in RGC axonal growth cones”.

Translational targets for Netrin-1 and Sema3A have been described by the authors and others but are being ignored in this manuscript. Are the mRNAs coding for β-actin, Par-3, Tctp associated with DCC, or is RhoA mRNA associated with Nrp-1? If they are not part of the interactomes, what does this mean for the proposed model?

We probed for the mRNAs listed by the reviewer:

- β-actin is enriched in DCC compared to Nrp1;

- TCTP is higher in DCC, although not significantly and it has a very low FPKM;

- Par-3 is not detected;

- RhoA is detected at very low FPKM and it is not significantly different.

As specified in the original manuscript, we used the human cell line SH-SY5Y rather than whole *Xenopus* brains to ensure that any detected differences in mRNA binding were not due to the expression of DCC and Nrp1 in different cell types. We were not able to use retinal axons because this would not generate sufficient material to perform pulldowns. Therefore, the receptor interactomes in this cell line are likely to partially differ from the receptor interactomes of the retinal axons where the particular proteins named by the reviewer are known to have key roles in axon guidance. This is exemplified by the absence of *rps14* mRNA enrichment in SH-SY5Y cells, which was instead detected in the *Xenopus* brain (Figure 4—figure supplement 1C). We validated, three mRNAs – *ctnnb1, hnrnph1* and *rps14* – to interact with DCC in the *Xenopus* brain (Figure 4D, Figure 4—figure supplement 1E), and we validated that their corresponding proteins are selectively increased in axons in response Netrin-1 but not Sema3A, consistent with our axonal proteome study (Cagnetta et al., 2018) and with the detection of their mRNAs in retinal axons (Shigeoka et al., 2018). These results point to a model where receptor-specific interactomes act as a hub for cue-induced selective translation. As mentioned in the Introduction of the original manuscript, there are several other non-mutually exclusive mechanisms that contribute to the cue-induced selective translation in axons, including microRNA regulation (Bellon et al., 2017), mRNA modification (Yu et al., 2018), modulation of the phosphorylation of eukaryotic initiation factors (Cagnetta et al., 2019), and RBP phosphorylation (Huttelmaier et al., 2005; Lepelletier et al., 2017; Sasaki et al., 2010).

Other major points in order of occurrence:Figure 1G-J: The presented western blots are single experiments without replication and are not quantified. The presentation of single, non-repeated experiments is a recurrent problem in this manuscript.

We apologize for causing this misunderstanding. These Western blots have been repeated at least 3 times, some of them many more times. We have now made this clear in the revised manuscript in the figure legends and Material and methods section.

Figure 1—figure supplement 1F, G: RPL39 is mentioned in the main text, but does not appear in the figure. The variances in this figure are quite large, and as the authors state, the conclusion is preliminary. The authors should either perform more experiments to elevate the conclusion from preliminary or remove this entire figure/chapter.

We only performed relative quantifications for RPs that were present in all three replicates of at least one of the pulldowns. RPL39 was not detected in the Nrp1 pulldowns and present in only 2 out of 3 replicates of the DCC pulldown (as mentioned in the original manuscript) and did therefore not appear in the figure.

We first tried to validate the differential association of two RPs (RPL39 and RPL4) by IP-Western blot but the antibodies were not suitable for Western blot on *Xenopus laevis* samples as they generated non-specific bands. We then tried to strengthen this point by testing whether knocking down one of the differentially enriched ribosomal proteins affects the cue-specific selective translation. Specifically, we have performed knockdown of RPL4 and RPL39 with morpholinos. Unfortunately, these results show that while RPL39 morpholino does decrease RPL39 levels (23% decrease), it also decreases the levels of two other ribosomal proteins we tested (RPS7 – 28% decrease and RPL19 – 27% decrease) in axonal growth cones. This result makes it impossible to use this approach for testing the effect of differential ribosomal proteins on cue-induced selective translation. In addition, the RPL4 morpholino that we tested did not result in a knockdown of RPL4 in axonal growth cones. Since we are unable to strengthen our conclusions, we have, as the reviewer suggests, now removed this data from the manuscript. It is important to note that these results are not necessary for the main conclusions of our study.

Figures 2—figure supplement 1A, B: Single, unrepeated experiments without a proper quantification. This is especially troublesome, as the purported shift for DCC upon EDTA treatment is not obvious at all.

We apologize again for not including this information about replication in the original manuscript. We have replicated and quantified these fractionations for DCC. We also quantified the experiments we have (Figure 2—figure supplement 1C). Our results are consistent with those from Tcherkezian et al., 2010, where they also performed polysome fractionation with and without EDTA for the DCC receptors. Furthermore, the results shown in Figure 2A-C strongly support that EDTA treatment results in loss of association of receptors to ribosomes, consistent with these polysome experiments. Importantly, we performed these experiments to provide further evidence of the association of receptors with ribosomes detected using several different approaches: i) IP followed by Western blot and mass spectrometry (Figure 1A-F), ii) IP followed by qPCR (Figure 1G-J), iii) expansion microscopy imaging (Figure 3A-B), iv) Proximity Ligation Assay (Figure 3C-D).

We added the new data in Figure 2—figure supplement 1C and related text in the subsection “Guidance cue receptors associate with ribosomes in a mRNA-dependent manner”.

Figure 2A: The RNAseA/T1 treatment is an elegant approach but to follow the authors interpretation it would be critical to experimentally prove that the rRNAs stay intact.

We agree that it is critical to show that rRNA is not affected by our RNAseA/T1 treatment. We have now provided new data from sucrose density gradients with and without RNAseA/T1 treatment. The UV absorbance profiles clearly show that unprotected mRNAs are cleaved, resulting in a loss of polysomes. The monosomal peak clearly increases indicating no or minimal loss of ribosomes/ribosomal RNA during our treatment. Prolonging the treatment to 30 min (instead of the 15 minutes that we used in our experiments) shows the same result. We have now added these new data in Figure 2—figure supplement 1D and related text in the subsection “Guidance cue receptors associate with ribosomes in a mRNA-dependent manner”.

Figures 2D, E: The author present the same plots as in Figure 1A without mentioning this fact. Also, there is no congruency between the main text, the figure legend and the actual figure: where are the 10 shared RBPs?

We thank the reviewer for this feedback. We had highlighted the ribosomal proteins (RPs) in Figure 1 and the RNA binding proteins (RBPs) in Figure 2. We have now replaced these panels with a new heatmap analysis of all the RBPs pulled-down after DCC and Nrp1 pulldown (now Figure 2D). This new heatmap provides a clearer overview of which RBPs are pulled-down and which of these are specific or common between the two receptors.

Figure 2—figure supplement 1C, D: Single unrepeated, non-quantified experiments.

These IP-Western blot experiments were repeated two times and the mass spectrometry experiments, which were also repeated 3 times, showed consistent results for this RBP (Figure 2D). Furthermore, we have now replaced this figure with new, more informative data performed in axonal growth cones. In this new experiment, we have quantified the co-localization between DCC, Nrp1 and the RBPs Staufen1 and hnRNPA2B1. The results show that DCC co-localizes to a significantly higher degree with hnRNPA2B1 compared to Staufen1 and, conversely, that Nrp1 co-localizes to a significantly higher degree with Staufen1 compared to hnRNPA2B1. These results support the preferential or selective binding of specific receptors to specific RBPs. We have now added this data in Figure 2E-F and related text in the subsection “DCC and Nrp1 bind to specific RNA-binding proteins”.

Figures 3A, B: Single growth cones are presented without replication, quantification or any unbiased co-localization analysis.

We performed these experiments on 4 independent biological replicates for DCC (total growth cones = 73) and 4 independent biological replicates for Nrp1 (total growth cones = 72). This information has now been added to the figure legend of Figure 3—figure supplement 1. We have quantified the co-localization in unstimulated conditions. We have performed unbiased co-localization analysis and found a Pearson’s correlation of 0.432 and Mander’s overlap coefficient of 0.434 for DCC and RPL5 and a Pearson’s correlation of 0.673 and Mander’s overlap coefficient of 0.675 for Nrp1 and RPS3a, confirming the partial co-localization claimed in the original manuscript. We have now added the Pearson’s correlation values to Figure 3—figure supplement 1A and related text in the subsection “Receptor-ribosome coupling occurs in RGC axonal growth cones” and have added the number of biological replicates to the corresponding figure legend.

Figure 3E: Why did the authors normalize the data instead of presenting the actual counts per sq. mm?

To ensure the highest quality of analysis, we always normalize each experimental replicate to the control condition and finally pool the normalized replicates together.

After having performed many PLA experiments, we have noticed that the absolute counts of PLA signal sometimes differ between experiments and between the use of different kits. To account for these differences, we therefore normalized our PLA data.

In these specific experiments, the average absolute count per growth cone for EphB2-RPL5 was equal to 0.54, with 75% of growth cones showing no PLA signal. By contrast, the average absolute count per growth cone for DCC-RPL5 was equal to 7.31 counts per growth cones, with less than 5% of growth cones without PLA signal. The average absolute count per growth cones for Nrp1-RPS23 was equal to 2.14, with less than 25% of growth cones without PLA signal.

Figure 3F: The authors are correct by stating that association of DCC with RPL5 and RPS4X is significantly decreased, however, they ignore that vastly different effect sizes. RPL5 seems reduced by ~80% while RPS4X is reduced by less than 20%. What is the explanation for this and shouldn't the reduction of the ribosomal subunits be stoichiometrically matched to form functioning ribosomes? The authors argue that local production of RPLs might obscure their results; they should perform the experiments in the presence of CHX to excluded this potentially confounding possibility.

The reviewer is correct, we know that both RPL5 and RPS4X are locally translated in response to Netrin-1 (Cagnetta et al., 2018; Shigeoka et al., 2018), suggesting that the significant decrease in PLA signal detected is even underestimated. Therefore, we think that repeating these experiments with CHX would not provide significant new information. In addition, it should be noted that in other experiments (Figure 4A and new Figure 3L), the Netrin-1-induced decrease in PLA signal of DCC-RPL5 is more in line with the Netrin-1-induced decrease in DCC-RPS4X PLA signal and with the Sema3A-induced decrease in the Nrp1-RPS3A and Nrp1-RPS23 PLA signals. It is possible that some variability may be due to the company selling the PLA kits changing hands during the course of this study. What is key is that in all these experiments there is a significant decrease in the PLA signal for all the RPs tested. Additionally, we have performed new strengthening control experiments showing that Sema3A does not affect the DCC-RPL5 PLA signal and Netrin-1 does not affect the Nrp1-RPS23 PLA signal (Figure 3J-K), thus corroborating the cue-specificity of the decrease in the receptor-RP PLA signal.

These new data have been added in Figure 3J-K and related text in the third paragraph of the subsection “Dissociation of ribosomes from receptors is triggered by extrinsic cues and requires endocytosis”.

Figure 4A: It is surprising to see that Netrin-1 does not induce local translation as measured by puromycylation. How do the author reconcile this finding with previous reports that Netrin-1 induces local translation?

Down-regulation of global translation does not exclude the selective translation of a subset of mRNAs. This is exemplified by the stress response, where, despite a strong decrease in global translation, ~5% of the genome is selectively up-regulated. In our culture conditions, where profuse axon growth is obtained on high laminin substrate, Netrin-1 acts as a repulsive cue (Hopker et al., 1999) and, although repulsive Netrin-1 down-regulates axonal translation (Figure 4B-C), the local synthesis of a subset of proteins is simultaneously up-regulated (Cagnetta et al., 2018). We have now clarified this point in the subsection “Integration of multiple cues can affect the cue-induced selective translation of receptor-specific mRNAs”.

Figure 5A: A single image without biological repeats or (unbiased) quantification.

This is a representative EM image of many that we obtained from 22 different growth cones. We have now included the n numbers and additional EM images, as well as new quantitative analyses in Figure 3F-G and Figure 3—figure supplement 1C-E and in the subsection “Receptor-ribosome coupling occurs in RGC axonal growth cones”. These analyses shows that 20 out of 22 axonal growth cones contain rows of ribosomes aligned just under the plasma membrane (within 50 nm) and the remaining two growth cones contain single ‘isolated’ ribosomes under the plasma membrane. In addition, the inter-ribosome distance measurements revealed, interestingly, that the ribosomes are spaced significantly further apart in growth cones than in cell somas. This is consistent with single ribosomes, monosomes, binding to the intracellular portions of transmembrane receptors, such as DCC or Nrp-1. We have added these new data and EM images in Figure 3F, 3G and Figure 3—figure supplement 1C-E and related text in the aforementioned subsection.

We included this EM image because it demonstrates the striking spatial arrangement of ribosomes lined-up under the plasma membrane in growth cones. We have now moved it to earlier in the manuscript to Figure 3, to clarify that the EM data provide a “snap-shot” view further confirming that ribosomes lie in close proximity to the plasma membrane.

Reviewer #3:[…]1) The differential association of specific RPs with DCC vs. Neuropilin-1 is intriguing and distinct ribosome populations have been suggested in a number of systems but unequivocally proven. Some of the techniques used here and methodologies rely on those in the Shigeoka et al., 2018 reference from the authors. Particularly, the experiments with RNAseA/T1 in Figure 2. The reference there is incomplete and I do not see it has been published except for bioRxiv (I am not certain on eLife's policy for referencing unreviewed work like this).

We agree that it is crucial to show that our experimental treatments are working effectively. To provide independent evidence that this is the case, we have performed polysome fractionations and the resulting profiles show that RNAseA/T1 treatment decreases polysomes, whereas monosomes and subunits stay intact, as expected (now Figure 2—figure supplement 1D). We have additionally cited another article that uses similar conditions (Simsek et al., 2017).

These new data are added in Figure 2—figure supplement 1D and related text in the subsection “Guidance cue receptors associate with ribosomes in a mRNA-dependent manner”.

2) Assuming the RNAseA/T1 treatment effectively shows that mRNA/protein interactions are needed for the ribosome subunits to interact with DCC and Nrp1, it is surprising that no efforts are made to determine whether RNA is needed for the RBP associations with these receptors. Similarly whether ribosome subunits association is needed for mRNA interaction with DCC and Nrp1.

These are interesting questions. We have added new data using Western blot analysis in which we quantified the effect of RNAseA/T1 treatment on the interaction of the RBP, Staufen1, with Nrp1 (new Figure 2—figure supplement 1E). This shows a small but significant decrease in Staufen1 binding to Nrp1. This result suggests that mRNA may partly stabilize the interaction between receptors and RBPs. In addition, we have performed new experiments that show cue stimulation (with Netrin-1) did not result in a decrease in DCC-hnRNPA2B1 PLA signal, indicating that RBPs remain attached to receptors after cue stimulation (new Figure 3—figure supplement 1F).

The second question has been more difficult to answer. We performed new qPCR experiments after DCC pulldown in the presence/absence of EDTA and the results indicated that ribosome subunit association does affect the interaction of β-actin mRNA with DCC. However, because EDTA treatment may have other effects in this situation, we are not sufficiently confident to include the data in the manuscript.

We have now added the new data in Figure 2—figure supplement 1E and related text in the subsection “DCC and Nrp1 bind to specific RNA-binding proteins” and Figure 3—figure supplement 1F and related text in the subsection “Dissociation of ribosomes from receptors is triggered by extrinsic cues and requires endocytosis”.

3) The MS data for different RPs in Figure 1—figure supplement 1 is surprising and does support the authors' suggestion for different ribosome populations. It is surprising that the authors did not validate those showing greatest differentials (e.g., L4, L35, S6 and S9). Also, it seems like there are more 60S than 40S RPs showing differential association, but the authors did not comment on this point.

This concern was also raised by the other reviewers, and as we said in response to them, we attempted to strengthen the findings by IP-Western blot quantification, based on the antibodies tested in *Xenopus laevis* available in our lab. However, the antibodies against RPL39 and RPL4 generated non-specific bands and were not suitable for Western blot on *Xenopus laevis* samples. We therefore attempted a functional validation by performing morpholino based KD experiments against two differential RPs (RPL39 and RPL4) in axonal growth cones in order to test their possible role in selective translation. These results show that RPL39 KD does decrease RPL39 levels (23% decrease) but also decreases the levels of two other ribosomal proteins (RPS7 – 28% decrease and RPL19 – 27% decrease) in axonal growth cones. This makes it impossible to use this approach for testing the role of a specific RP on cue-induced selective translation. Morpholinos to RPL4 did not knock down the protein in axonal growth cones. Therefore, as it has not been possible for us to independently verify, either by Western blot or function, the differential association of specific RPs with specific receptors, we have eliminated this point from the manuscript as the other reviewers suggested.

We thank the reviewer for pointing our attention to the difference in 60S and 40S RPs that showed differential association. Based on the current literature on heterogeneous/specialized ribosomes, there does not seem to be a bias towards 60S RPs in heterogenous ribosomes. We can only speculate as to why we detect more 60S RPs to be differential but, since we could not validate our findings, we have removed these data and the accompanying discussion.

4) For Figure 2—figure supplement 1A-B, the majority of Nrp1 and DCC signals do not fractionate with subunits. Granted, these are very hard experiments and the EDTA treatments do shifts signals for Nrp1 and DCC upwards in the gradients. I think this would be strengthened EDTA treatments and some quantitation for the colocalization data shown in Figure 3 and Figure 3—figure supplement 1.

We appreciate the acknowledgement that these are not easy experiments to perform. We presented this data to provide further support for the association of receptors with ribosomes, in addition to the other experimental approaches that we used: i) IP followed by Western blot and mass spectrometry (Figure 1A-F), ii) IP followed by qPCR (Figure 1G-J), iii) expansion microscopy imaging (Figure 3A-B), iv) proximity ligation assay (PLA) (Figure 3C-D). Control and EDTA treated samples were compared directly, twice for DCC and once for Nrp1. We have quantified the relative amounts of DCC (n = 2 for control and EDTA) and Nrp1 (n = 2 for control, n = 1 for EDTA) in ribosome-free and ribosome-containing fractions. This relative quantification corrects for any possible decreases in overall protein levels, showing a clear shift to lighter fractions (i.e. ribosome-free fractions) for both DCC and Nrp1 after EDTA treatment (new Figure 2—figure supplement 1C). It is noteworthy that our results are consistent with those from Tcherkezian et al., 2010, which performed similar experiments for DCC. Furthermore, the results shown in Figure 2A-C strongly support that EDTA treatment results in loss of association of receptors to ribosomes, consistent with the polysome experiments.

We have obtained many expansion microscopy images from four biological replicates and we have added the quantification of co-localization (new Figure 3—figure supplement 1A).

The new data are added in Figure 2—figure supplement 1C and Figure 3—figure supplement 1A and related text in the subsections “Guidance cue receptors associate with ribosomes in a mRNA-dependent manner” and “Receptor-ribosome coupling occurs in RGC axonal growth cones”.

5) It is surprising that no colocalization for mRNAs with the receptors is provided beyond the co-IP, and such RIP experiments are known to have artefacts from in solution RNA protein-interactions after lysis.

In this study we have reported that replicates of IP for DCC and Nrp1, followed by RNA-seq, show distinct but consistent signatures of mRNAs. It is technically very challenging to test the co-localization of proteins with specific mRNAs as mRNAs may be masked by the presence of binding proteins (Buxbaum et al., 2014). Furthermore, for any specific mRNA we have tested previously with in situ hybridization, or live, there are rarely more than 8-15 fluorescent mRNA puncta per growth cone. By contrast, DCC and Nrp1 receptor immunostaining gives an abundant signal throughout, with hundreds of fluorescent puncta per growth cone. Therefore, we think that co-localization of receptors with specific mRNA probes (FISH) would not be a convincing approach.

Since mRNAs bind to RBPs and these are likely essential for the mRNA-receptor association, we reasoned that a more feasible and equally informative experiment to address this question would be to test the co-localisation of RBPs with receptors. To this end, we performed dual immunocytochemistry and analysed the co-localization of DCC and Nrp1 with hnRNPA2B1 and Staufen1. The new data show that DCC co-localizes to a higher degree with hnRNPA2B1 compared to Staufen1 and that Nrp1 co-localizes more with Staufen1 compared to hnRNPA2B1. This is in line with, and validates, the differential binding results of our co-IP data.

We have added these data in Figure 2E-F and related text in the subsection “DCC and Nrp1 bind to specific RNA-binding proteins”.

6) For Figure 5A, the EM is compelling an shows a surprisingly high number of electron dense ribosome like structures. The authors do not note whether this is a stimulated or naïve culture. Comparison of the two would be informative and strengthen the authors' conclusions (though I do recognize the difficulty in this request).

We apologise for the lack of clarity. The EM experiments were carried out on unstimulated growth cones. We have now specified this in the subsection “Receptor-ribosome coupling occurs in RGC axonal growth cones”, and the figure legend of Figure 3.

Although it would be nice to show stimulated growth cones by EM too, and compare them quantitatively, we think that an ultrastructural analysis of this sort would take many months to complete and would be a major study in itself. Here we aimed to provide further visual evidence that ribosomes can be found in close proximity to the plasma membrane in axonal growth cones. This finding has been extensively corroborated by the several receptor-ribosome proximity-based (PLA) experiments, quantified and in triplicate, together with negative controls (Figure 3C-E, H-I and Figure 3—figure supplement 1B, H), in stimulated versus naïve conditions. Furthermore, we have now added new strengthening control experiments showing that Sema3A does not affect the DCC-RPL5 PLA signal and Netrin-1 does not affect the Nrp1-RPS23 PLA signal, confirming the cue-specificity of the decrease in the receptor-RP PLA signal (Figure 3J-K and related text in the third paragraph of the subsection “Dissociation of ribosomes from receptors is triggered by extrinsic cues and requires endocytosis”).

To strengthen our EM findings further, we have now added additional images, as well as new analysis of these EM images in Figure 3F-G and Figure 3—figure supplement 1C-E. These analyses show that 20 out of 22 axonal growth cones contain rows of ribosomes aligned just under the plasma membrane (within 50 nm) and that the remaining two growth cones contain single ‘isolated’ ribosomes under the plasma membrane. Interestingly, we observed that the inter-ribosome distance in these growth cone ribosomes is significantly larger than ribosomes in the cell soma. This is consistent with single ribosomes, monosomes, binding to the intracellular portions of transmembrane receptors, such as DCC or Nrp-1. We have added these new data and analyses in Figure 3F, 3G and Figure 3—figure supplement 1C-E and related text in the subsection “Receptor-ribosome coupling occurs in RGC axonal growth cones”.

7) Robo3.3 and EphA2 have been shown to be translated in axons. This is probably worth mentioning in the Discussion since the authors show that EphB2 does not use this mechanism for sequestering RNPs and ribosome-bound mRNAs.

This point is unclear as we think that the lack of association of a receptor with ribosomal proteins does not exclude that the receptor itself may be potentially locally translated.

[Editors’ note: the author responses to the re-review follow.]

[…] Two of the reviewers found your manuscript satisfactorily amended, but one of the reviewers cited a number of points that should be addressed before publication, all of them addressed by textual amendments:1) There are a number of places where your results are somewhat overstated and should be toned down:a) Abstract: 'Our findings.… provide a general model for the rapid, localized and selective control of cue-induced translation.' Compare this to the much more accurate and measured summary: '… this study provides evidence…and suggests…'. Your results are interesting and represent a significant advance, but it may be premature to consider that they comprise a general model.

We have now changed the Abstract, stating:

“Our findings reveal receptor-specific interactomes and suggest a generalizable model

for cue-selective control of the local proteome”.

b) The findings from Figure 2B, C in are summarized: '…these results suggest that the interaction.… is likely mediated through mRNA'. One page later, this sentence has morphed into the definitive statement: '…our finding that mRNA mediates the association of receptors with specific ribosomes…'

We understand the reason for this comment, but the results of the paper, as it has been revised with new data, are stronger with respect to this point. We have therefore changed the statement into the following sentence:

“Together with our evidence implicating mRNA in the association of receptors with ribosomes, these results are consistent with a model in which receptors associate with specific RBPs, which bind specific mRNAs, and these mRNAs, in turn, recruit ribosomes.”

c) '…reveal that multiple receptors.… can associate with ribosomes.' This statement is imprecise; the results up to this point show association with ribosomal subunits, not ribosomes.

We agree with the reviewer that it is more accurate to describe it is an association with ribosomal subunits and have now adapted this in the revised manuscript: “…reveal that multiple receptors.… can associate with ribosomal subunits.”

Please carefully edit the Results section and relegate the interpretation especially statements regarding the general impact/significance, to the Discussion.

We have done this.

2) A major concern in the first review was that some experiments were not repeated and quantified. Most of these issues have been fixed, but in Figure 1C-E, Figure 1 —figure supplement 1A-C, results are still presented without quantification. The sentence in the figure legend that these experiments have been repeated three times but the actual results should be presented.

We have now quantified the protein bands in the IP samples relative to the input by densitometry analysis using ImageJ software on the co-IP-Western blot results presented in Figures 1C-F and Figure 1—figure supplement 1A-C (see Author response image 1). These co-IP-Western blot experiments were done to validate whether or not ribosomal proteins bind to the different receptors in a simple ‘yes’ or ‘no’ assay. In general, co-IP-Western blots are used to test whether one protein binds another. Binding is confirmed by the existence of a band of the appropriate molecular size and antigenicity. The main result here, which is clear in the blots presented, and also in the graphs shown below, is that DCC, Nrp1 and Robo2, but not EphB2, bind to ribosomal proteins. These IP results support the mass spectrometry data (Figure 1A, B) and IP-qPCR data (Figure 1G-J and Figure 1—figure supplement 1D, E). These two other techniques provide useful quantitative information about protein levels and interaction with ribosomal subunits (i.e. ribosomal RNA). However, the quantifications below on the co-IP-Western blots do not provide useful information about quantities of protein as replicates were ran on different gels and many different factors influence the density of the bands (e.g. transfer efficiency, antibody incubation time, antibody lot numbers, development time). As quantification of these results is not informative other than providing some assurance that the bands are real, we would prefer not to include these graphs in the revised manuscript, but provide them here, in response to the comment above, to show that the results are robust.

3) Figure 2—figure supplement 1C: the n for these experiments are 2 or 1. Showing bar graphs with SDs is not very meaningful for such low n numbers. The individual data points should be shown instead.

We agree that presenting the individual data points is more informative. We have therefore replaced the bar graphs in Figure 2—figure supplement 1C with a graph showing the individual data points.

4) Subsection “DCC and Nrp1 bind to specific subsets of mRNAs”, first paragraph: The finding that high abundance mRNAs are much more differentially associated than low abundance mRNAs seems to indicate that these mRNAs are the mRNAs that conform to the authors proposed model. It is a missed opportunity to not talk more about these mRNAs: provide numbers, identity, GO analysis.

We thank the reviewer for this comment. Indeed, GO analysis of high abundance mRNAs (FPKM >100) reveals specific GO categories associated with Nrp1 and DCC, such as ‘cell-cell adhesion’ and ‘protein targeting’ for DCC and ‘translation’ and ‘small GTPase mediated signal transduction’ for Nrp1. We have now added the results in Figure 2—source data 2. This supplementary table also contains the numbers and identity of all differentially expressed genes (DEG) and high abundant genes with an FKPM >100 are highlighted.

In addition, we have added a discussion of these results to the subsection “DCC and Nrp1 bind to specific subsets of mRNAs”.

5) In response to conceptual question 2 (reviewer 2, first review), you provide the interesting finding that endocytosis is required for the dissociation between DCC and RP. However, the question was in relation to the mRNA-seq findings: it is my understanding that the RNA-seq experiments cannot distinguish between mRNAs associated with the receptors at the membrane vs. receptors at vesicles. For example, if mRNAs/RNPs are associated receptors on vesicles that transport the receptors into axons, these mRNAs would show up in the RNA-seq, even if they no longer associate with receptors after insertion into the plasma membrane. Please discuss this point to make clear that the RNA-seq results might include mRNAs that are not associated with receptors in the membrane.

The reviewer is correct to say that we cannot determine where exactly the association of receptors with mRNAs takes place from the RNA-seq experiments. It is indeed possible that some of the detected mRNAs are associated with receptors while traveling on endocytic vesicles and we have now discussed this possibility in the fourth paragraph of the Discussion.

6) The response to the comment regarding Figure 3F is unsatisfactory: You seem to indicate that the data as presented are unreliable due to problems with the reagents. Expand on this point.

The data presented in Figure 3F (Figure 3F in the initial submission, now Figure 3H) are reliable. They were obtained from three independent biological replicates with a total of more than a hundred growth cones per condition. There were no problems with the reagents or PLA signal in these experiments. However, as we indicated, the sensitivity of the technique varies between trials, especially between trials performed with kits from DuoLink and trials with the kits that were provided by Sigma-Aldrich. The DCC-RPL5 data in Figure 3H was performed with a PLA kit from the company Duolink before they were taken over by Sigma-Aldrich. All other experiments for DCC (Figures 3H – DCC/RPS4X, 3J, 3L and Figure 4A) were performed after this takeover and showed a smaller decrease in PLA signal after Netrin-1 stimulation compared to the first set of experiments. We are unsure as to what exactly caused this difference, but it co-occurred with the change of companies. What is most important in these experiments is that there is a significant decrease in PLA signal after Netrin-1 stimulation for all tested RPs, whereas Sema3A stimulation does not affect DCC-RPL5 PLA (Figure 3J), and that this is true with both the DuoLink and the Sigma-Aldrich kit. Indeed, although the sensitivity between kits may be different, the experiments are all internally consistent and are always done with appropriate controls. So the results are even perhaps more reliable simply because they were reproducible with different kits.